# Insight into the Characterization of Volatile Compounds in Smoke-Flavored Sea Bass (*Lateolabrax maculatus*) during Processing via HS-SPME-GC-MS and HS-GC-IMS

**DOI:** 10.3390/foods11172614

**Published:** 2022-08-29

**Authors:** Hua Feng, Vaileth Timira, Jinlong Zhao, Hong Lin, Hao Wang, Zhenxing Li

**Affiliations:** College of Food Science and Engineering, Ocean University of China, No. 5, Yushan Road, Qingdao 266003, China

**Keywords:** smoke-flavored sea bass, volatile compounds, HS-SPME-GC-MS, HS-GC-IMS, smoke flavoring

## Abstract

The present study aimed to ascertain how the volatile compounds changed throughout various processing steps when producing a smoke-flavored sea bass (*Lateolabrax maculatus*). The volatile compounds in different production steps were characterized by headspace-solid phase microextraction-gas chromatography-mass spectrometry (HS-SPME-GC-MS) and headspace-gas chromatography-ion mobility spectrometry (HS-GC-IMS). A total of 85 compounds were identified, and 25 compounds that may be considered as potential key compounds were screened by principal component analysis (PCA) and partial least squares discriminant analysis (PLS-DA). Results indicated that aldehydes were the major volatile compounds throughout the processing. The characteristic volatile compound in fresh samples was hexanol, and curing was an effective method to remove the fishy flavor. The concentration of volatile compounds was significantly higher in dried, smoked, and heated samples than in fresh and salted samples. Aldehydes accumulated because of the drying process, especially heptanal and hexanal. Smoke flavoring was an important stage in imparting smoked flavor, where phenols, furans and ketones were enriched, and heating leads to the breakdown of aldehydes and alcohols. This study will provide a theoretical basis for improving the quality of smoke-flavored sea bass products in the future.

## 1. Introduction

Among fish products, smoked fish is a processed food with high economic value. Traditional smoking is a combination of salting, drying, hot or cold smoking processes; this process can extend the storage period of fish while also giving it a distinctive smoky flavor that will appeal to more consumers. However, the direct contact between fish and smoke in the traditional smoking process brings potential hazards and safety issues [1], so smoke flavorings have been developed as a successful alternative to traditional smoking to obtain a smoky effect [2]. According to the survey by Alçiçek et al. [3], the United States (75% of the market) and Europe (30% of the market) are the largest consumers of liquid smoke. The global market size of liquid smoke reached USD 56.5 M in 2018 [4]. Sea bass (*Lateolabrax maculatus*) is one of the most important economic fish in East Asia [5], ranking third among the mariculture species in mainland China, with an annual production of more than 150,000 t [6]. Sea bass has demonstrated good attitude toward smoking and is a good alternative to traditional smoked fish, such as salmon or trout [7]. The most significant determinant of the overall quality of smoked sea bass for consumers is its smoked flavor. Therefore, the production of smoke-flavored sea bass with desirable flavor is crucial for both producers and researchers of smoked fish.

The quality of smoke-flavored sea bass and the formation of its flavor is related to the characteristics of raw fish and production operations, such as the variability of raw materials (fat content) [8,9], brine concentration [10], and smoking techniques. The flavor formation of smoke-flavored sea bass is related to the production process. The traditional smoking process in which the smoke and fish are in contact for a long time produces a thick smoky flavor, while the modern smoke flavoring is affected by factors such as short smoking time and unstable smoke flavors, resulting in a lack of flavor in smoked product [11,12]. Nithin et al. [13] produced masmin flakes by smoke flavor, which could obtain products that matched the taste of traditional masmin. Ruiz et al. [14] optimized the “smoke flavoring” process to achieve the best sensory properties in smoke-flavored tilapia fillets. While other researchers have carried out similar research, their studies have largely focused on sensory aspects rather than the flavor-related production pathways for volatile compounds in products [12,15,16]. Currently, instruments are available to analyze flavor compounds, and these techniques could counteract the non-objective judgment of sensory evaluation. Flavor analysis techniques have been applied to the identification and separation of volatile substances in foods such as fermented squid [17], sauce spareribs [18], and shrimp [19], and researchers have explored the changes of volatile substances in cold-smoked Spanish mackerel [20] and fermented fish [21] during processing. The detail of the improvement of HS-SPME-GC-MS for the qualitative and quantitative analysis of volatile compounds is still needed [22]. However, it requires tedious pre-processing and long detection time, which limits the efficacy of GC-MS [23]. HS-GC-IMS has been widely used to analyze volatile substances in food [24,25,26,27], because of the separation feature of GC and the high sensitivity of IMS. GC-IMS does not need the pre-treatments and displays the results of the analysis in a color contours image, making it possible to visually display differences among samples [28]. In addition to instrumental analysis, statistical analysis can be used to distinguish characteristic volatiles between different variables [29,30]. For example, LEE et al. [31] distinguished the characteristic metabolites in each fermentation step of soybean paste by multivariate statistics. Therefore, the combined use of multiple flavor analysis techniques could provide more comprehensive and accurate analytical results [32,33].

Hence, this work investigated the changes of volatile compounds in smoked sea bass at different stages of processing. The diversity of volatile substances in smoke-flavored sea bass was obtained by using a combination of HS-SPME-GC-MS and HS-GC-IMS. Furthermore, PCA and PLS-DA were performed to elucidate the correlation between volatile compounds and different processing stages. Overall, these studies may help to further improve the smoking technique (with fish cooking) and enhance the flavor quality of smoke-flavored fish.

## 2. Materials and Methods

### 2.1. Smoke-Flavored Sea Bass Preparation

The process used to produce smoke-flavored sea bass is shown in Figure 1 [34,35]. Fresh sea bass purchased from a local market (Qingdao, China) was transported live to the laboratory. The average weight of the fish was 534.5 g (±32.8 g). On a sterile operating table, the fresh sea bass was slaughtered, headed, degutted and washed. Then, it was cut into equal-sized fillets (10 × 5 × 3 cm) for processing after washing. It should be noted that one side of the sea bass fillets was with skin. The average weight of all the fillets was 110.5 g (±8.5 g). The salt used for the salting stage was acquired from a local supermarket, and the smoke flavors used for the smoke flavoring was provided by the Shuanghui Food Co. (Luohe, China) and were water-soluble natural liquid smoke flavoring (“SMOKEZ ENVIRO 24 PB”, Red Arrow International LLC, Manitowoc, WI, USA) with a pH of 3.74.

Smoked-flavored sea bass processing method: the pre-treated fillets were immersed in a 10% salting solution (1:1, *w/v*) for 4 h; then, they were dried in a hot-air drying oven (60 °C, 2 h) to reach 65% (± 5%) of the original weight by dry weight; immersed in a solution of smoke flavoring in water (1:5, *v/v*) for 2 min with a fish-to-flavoring solution proportion of 1:15 (*w/v*), and smoked fillets were kept at room temperature for 2 h to reacting between smoke components and fish flesh and then heated by heating in an oven (180 °C, 10 min).

Samples of sea bass were collected for analysis after five of the key stages in smoke-flavored sea bass production (Figure 1). Three fillets were randomly selected at each sampling stage; fish tissues from the same parts were chosen as the samples to be tested and vacuum-packed, and the samples of each sampling stage were numbered to indicate the three samples of each stage, which were frozen and preserved until needed for analysis.

### 2.2. HS-SPME-GC-MS Analysis

#### 2.2.1. Extraction of Volatile Compounds

Four grams of samples from different sampling points were transferred into a headspace vial (20 mL). Then, 40 μL 2,4,6-trimethyl pyridine (TMP, 10 PPM, internal standard (IS)) was loaded onto the head-space vial. The mixture was then balanced at 50 °C, for 30 min, and a 50/30 μm SPME fiber (DVB/CAR/PDMS, Supelco, Bellefonte, PA, USA) was inserted in the headspace vial. Be careful that the fibers did not touch the sample. The fiber was exposed to the headspace of the vial for extraction at 50 °C for 30 min and transferred to the injection port of the GC instrument (250 °C) for 5 min.

#### 2.2.2. HS-SPME-GC-MS Analysis of Volatile Compounds

Volatile compounds were measured by an HS-SPME-GC-MS system (8890/7000D, Agilent Technologies, Santa Clara, CA, USA) with an Agilent 5975C mass selective detector. An Agilent HP-5MS column (60 m × 0.25 mm × 0.25 μm) was used for the separation of volatile compounds. The heating procedure of the GC oven was as follows: 35 °C for 3 min, heated to 65 °C at 3 °C/min, heated to 180 °C at 8 °C/min, then heated to 200 °C at 15 °C/min and finally heated to 260 °C at 20 °C/min for 5 min. Helium was used as a carrier gas at a flow rate of 1.0 mL/min. Mass selective detection was performed in scan mode (m/z 45–550, EI (70 eV), ion source temperature 230 °C, quadrupole temperature 150 °C, transmission line temperature 280 °C) [36].

The results of the experimental mass spectra were compared with the mass spectra library from NIST11s, and the identification of volatile compounds was completed via comparing the retention indices (RI), which were calculated by analyzing n-alkanes (C7–C30) under the same chromatographic conditions and comparing with the NIST Standard Reference Database Number 69. Finally, the relative concentrations of the volatile compounds (semi-quantitative) were calculated by comparing the compounds with the IS (expressed as μg/kg).

### 2.3. HS-GC-IMS Analysis

Two grams of samples from different sampling points were transferred into a headspace vial (20 mL) [37], and the volatile components from all samples were identified by GC-IMS (Flavour Spec^®^, Dortmund, Germany). The GC-IMS program settings are shown in Table 1. Nitrogen was used as the carrier gas under the following programmed flow: 2 mL/min for 2 min, 15 mL/min for 8 min, 50 mL/min for 5 min, 100 mL/min for 5 min, 150 mL/min for 15 min, and then the flow stopped. The retention index (RI) was calculated regarding the standard (n-ketones C4-C8, Sinopharm Chemical Reagent Beijing Co., Ltd., Beijing, China), and volatile compounds were identified by RI and drift time of the standard in the GC-IMS library [38].

### 2.4. Statistical Analysis

Three parallel experiments were conducted for each sample, and the results were expressed as mean ± standard deviation (SD) (*n* = 3). Microsoft Office 2016 and Origin 2016 were used to draw and merge graphics. Heatmap analysis and PLS-DA were performed in MetaboAnalyst 5.0 (https://www.metaboanalyst.ca) (accessed on 12 August 2020). For HS-GC-IMS, samples were analyzed from different angles using the supporting Analytical software (LAV, Reporter plug-in, Gallery plot plug-in, GC-IMS Library search (G.A.S., Dortmund, Germany)). The analyses of variance (ANOVA) and Duncan’s means comparison test was applied with a significance level of 0.05 by using SPSS software (version 25, SPSS Inc. Chicago, IL, USA) [39].

## 3. Results

### 3.1. HS-SPME-GC-MS Analysis of Smoke-Flavored Sea Bass in Different Production Steps

#### 3.1.1. Identification of Volatile Compounds in Different Production Steps

The volatile compounds of smoke-flavored sea bass were tentatively identified by HS-SPME-GC-MS. According to Appendix A, out of 63 volatile compounds that were tentatively identified, sixteen phenols, ten aldehydes, ten alcohols, seven hydrocarbons, five ketones, four furans, three esters, three acids and five other compounds were mostly found in nine categories. There were different amounts of volatile compounds in each stage of processing, including sixteen in fresh fish (6 aldehydes, 3 alcohols, 2 acids, 5 hydrocarbons), ten in the salted sample (3 aldehydes, 1 alcohol, 1 acid, 5 hydrocarbons), sixteen in the dried sample (6 aldehydes, 2 alcohols, 2 acids, 6 hydrocarbons), forty in the smoked sample (5 aldehydes, 3 alcohols, 3 ketones, 15 phenols, 3 furans, 1 acid, 5 hydrocarbons, 5 other compounds), and thirty-seven in the heated sample (3 aldehydes, 5 alcohols, 4 ketones, 3 esters, 11 phenols, 2 furans, 6 hydrocarbons, 3 other substances).

#### 3.1.2. Changes in the Volatile Compounds in Different Production Steps

To further understand the differences in volatile flavor of smoke-flavored sea bass at different stages of processing, heatmap and hierarchical cluster analysis were used to visualize the data based on the concentrations of 63 volatile compounds initially identified (Figure 2), and analysis of ANOVA was conducted to analyze the initially identified volatile compounds.

In the cluster heat map, the distribution of dark red lattices was different, indicating that different samples have different characteristic volatile compounds. In the salted samples, the contents of hexanal, heptanal, nonanal, palmitic acid, undecane, dodecane, 1-octen-3-ol, tridecane and heptadecane decreased, and some alcohols (benzaldehyde, octanal, nonanal) and aldehydes (4-ethylcyclohexanol, 2-octen-1-ol, 2-ethylcyclohexanol) were not detected. The content of undecane, dodecane, tridecane, tetradecane, heptadecane, pristane, almitoleic acid, trans-2-undecen-1-ol and (Z)-7-hexadecenal were significantly higher in the dried samples. Aldehydes, ketones, phenols, and furans species all increased significantly after the fish was exposed to the smoke flavors, except for 15 phenolic compounds (4-ethylphenol, phenol, o-cresol, p-cresol, guaiacol, 2,4-dimethylphenol, 2-ethylphenol, 2-methoxy-5-methylphenol, 2,4,6-trimethylphenol, 4-propylphenol, 4-ethyl-2-methoxyphenol, 2,6-dimethoxyphenol, eugenol, dihydroeugenol, and ethylhydroquinone), and the other compounds all changed to different degrees. In the heated samples, fifteen volatile compounds (5-methyl furfural, 5-ethyl-2-furaldehyde, 3-furanmethanol, 4-methoxybenzhydrol, 3-methyl-2-cyclopenten-1-one, phenol, 2,4-dimethylphenol, 4-propylphenol, eugenol, ethylhydroquinone, 2-acetylfuran, 5-methyl-2-acetylfuran, palmitic acid, 3,5-dimethylpyrazole, and 3,4,5-trimethylpyrazole) disappeared and twelve new substances (1-nethylcyclohexanol, 2,3-dimethylcyclohexanol, 2-phenyl-2-norbornanol, geraniol, 2-methyl-2-cyclopentene-1-one, 4-hexen-3-one, methyl acetate, 2-methoxycarbonylimidazole, 1-octylformate, 3,5-dimethylphenol, 2-ethyl-5-methyl furan, and o-xylene) were formed.

To deeply explore the difference between volatile compounds in different production steps, the principal components were classified by PLS-DA score map (Figure 3a), and the total contribution rate of CP1 and CP2 reached 98.6%. Variable importance in projection (VIP) scores of various volatile compounds in smoke-flavored sea bass at various processing stages are shown in Figure 3b. The results show that there are eight substances with significant contribution (VIP > 1), including six phenols (guaiacol, o-cresol, 2-methoxy-5-methylphenol, p-cresol, 3, 5-dimethylphenol, and 4-ethyl-2-methoxy-phenol), one alcohol (3-furaldehyde) and one ketone (2-methyl-2-cyclopentene-1-one). These distinctive volatile compounds set the fresh, salted, dried, smoked, and heated samples apart to varying degrees.

### 3.2. HS-GC-IMS Analysis of Smoke-Flavored Sea Bass in Different Production Steps

#### 3.2.1. Identification of Volatile Compounds

HS-GC-IMS is a non-targeted analytical method for the identification of volatile compounds in a sample by providing retention time and drift time of the compounds. Figure 4 shows the 2D topographic subtraction plots of volatile compounds in smoke-flavored sea bass in different production steps. It was obtained by subtracting fresh fish stage from other samples. Each point represents a volatile compound, and the color and area of the point represent the size of the substance content. The redder the area is, the higher the content of volatile compounds, and the opposite is true for blue [40]. Figure 4 directly shows the migration time of each sample is about 13 ms, and the retention time of various volatile compounds is between 100 and 500 s. There were significant differences in volatile compounds at different stages. The content of some substances decreased in the salted stage, while the content of volatile substances increased significantly in the dried, smoked and heated samples, but there were multiple blue areas in the heated sample.

The smoke-flavored sea bass has a total of 26 volatile compounds that have been tentatively identified (Appendix A). Of note, the instrument detected seven volatile compounds (1-propanol, 2-propanol, 3-methylbutanol, 3-methylbutanal, hexanal, furfural and ethyl acetate) in both monomer and dimer forms [41]. Among the 26 volatile compounds tentatively identified, the carbon chains were generally concentrated in the range of C4–C9, which mainly included aldehydes, alcohols and ketones. There were 13 species of aldehydes, with the largest number, which was followed by alcohols (8) and ketones (5).

#### 3.2.2. Changes in the Volatile Compounds of Different Smoke-Flavored Sea Bass

To further analyze the effect of different processing stages on smoke-flavored sea bass, a fingerprint comparison of volatile compounds was conducted for each stage. Different columns represent various volatile compounds, and different rows represent samples at different stages (Figure 5). The color depth of signal points represents the concentration of the substance. As shown in Figure 5, the different stages show different fingerprint plots of volatile compounds. In region A, pentanal, acetic acid, 3-methylbutanal monomer, 2-methyl-1-propanol, ethyl acetate both monomer and dimer were present at each stage, while the contents of compounds except 3-methylbutanal dimer did not change significantly. Compounds in region B increased significantly during fish processing. They contain aldehydes such as butanal, ketones such as 3-hydroxy-2-butanone and alcohols such as propanol, which play an important role in the volatile compounds of smoke-flavored sea bass. The content of compounds in area C increased significantly in the dried sample, but these substances disappeared in the smoked sample. In area D, the content of compounds was reduced in the heated sample, including 3-pentanone, propanal, 3-methylbutanol. Volatile compounds in region E include 2-methylpropanal, furfural, 2-acetylfuran, and 2,3-butanedione, which were not found in the fresh, salted and dried samples, and the highest content was in smoked sample. With the increase of temperature, the content of these substances decreased. The characteristic volatile compounds of each stage help to distinguish smoke-flavored sea bass in different production steps.

PCA analysis could clearly highlight the volatile compounds differences in various processing stages based on the volatile compounds area signal intensity (Figure 6). The larger the sample distance in the figure, the more significant the difference between samples. The score chart shows that three principal components (PC1, PC2 and PC3) were obtained which accounted for 86.7% of the total variation. PC1, PC2, and PC3 explained 43.3%, 29.8%, and 13.6% of the variation, respectively. The distance between the samples in the heated and the others was significantly different, which indicated that volatile compounds in the heated sample was significantly different from other samples. Furthermore, the flavor curves of the dried and smoked samples were also different from those of other samples.

Aldehydes were the most prevalent volatile compounds in all phases of the samples, according to the classification diagram (Figure 7) of volatile compounds of smoke-flavored sea bass. The relative content of aldehydes was significantly higher in the dried and smoked samples than in the other groups (*p* < 0.05), and the concentration of aldehydes was significantly lower in the heated samples (*p* < 0.05). The relative contents of heptanal, 2,4-heptadienaland, hexanal, pentanal and 3-methylbutanal were the highest in the dried samples, while furfural and propanal were the highest in the smoked samples. The contents of 3-methylbutanal, furfural and propanal decreased significantly after high-temperature heating (*p* < 0.05). Alcohols were the lowest in the dried samples. The content of 3-methylbutanol monomer did not vary significantly (*p* < 0.05), while hexanol was highest in fresh samples and 1-propanol increased in dried samples. After treated with smoke flavors, the content of 2-propanol increased, but it significantly decreased in the heated samples (*p* < 0.05). Among the other volatile compounds, ethyl acetate and acetic acid did not change significantly (*p* < 0.05). The relative contents of 3-pentanone, pentane-2,3-dione, 3-hydroxy-2-butanone and 2-methylbutyric acid were significantly increased in the dried samples. The relative contents of 2,3-butanedione and 2-acetylfuran showed relatively high contents in the smoked samples. After high-temperature heating, the contents of 3-pentanone, 2,3-butanedione, and 2-acetylfuran decreased.

Additionally, ANOVA showed that 14 volatiles had notable differences between samples (*p* < 0.05). 1-Hexanol had the highest ionic signal intensity in fresh fish samples; heptanal, 2,4-heptadienal, hexanal, 3-pentanone, pentane-2,3-dione and 2-methylbutyric acid had the highest ion signal intensity in the dried samples; 3-methylbutanol, 2-propanol, furfural, 2,3-butanedione and 2-acetylfuran had the highest ion signal intensity in the smoked samples, and benzaldehyde and 2-methylpropanal had the highest ion signal intensities in the heated samples. By GC-IMS analysis, these 14 compounds may be considered potential key compounds.

## 4. Discussion

Samples of smoke-flavored sea bass were analyzed at different stages of the production line to determine where the key volatile substances may occur and which processing stage had the greatest impact on the flavor profile of the product. The combination of flavor analysis techniques could be used to better understand the formation of flavor in smoke-flavored sea bass and to control the production chain of smoke-flavored sea bass to improve the flavor quality of the product, which was important for the processing of smoked foods.

The diversity of volatile compounds in the different production steps was obtained using a combination of two instrumental analyses, and a total of 85 volatile compounds were initially identified, including 17 aldehydes, 16 phenols, 15 alcohols, 10 ketones, 4 esters, 4 furans, 5 acids, 7 hydrocarbons and 7 other compounds. Three aldehydes (heptanal, benzaldehyde and hexanal) and one furan (2-acetylfuran) had been identified via these two types of analysis methods, while a total of 22 volatile substances may be considered as potential key compounds. From the results, it was found that the two methods differed in their sensitivity to different classes of compounds, with most of the phenolic and heterocyclic compounds being detected by HS-SPME-GC-MS, while HS-GC-IMS detected compounds more inclined to aldehydes, alcohols, and ketones. Some compounds, including 2,4-heptadienal, 2-methyl-1-propanol, 2-methylpropanal, 3-hydroxy-2-butanone, 2-methylbutyric acid, etc., could only be detected by HS-GC-IMS. On the contrary, except for 2-isobutyl-3-methylpyrazine and 2-ethyl-5-methylpyrazine, other heterocycles could only be detected by HS-SPME-GC-MS. Thus, the combination of these two methods provides a more comprehensive aroma characterization [36].

The data [42] show that the volatile compound composition of fresh sea fish is comparable to that of freshwater fish, both of which are composed primarily of aldehydes and carbonyl and alcohol compounds. Most of them have flavor, although sea fish typically have a stronger flavor. HS-SPME-GC-MS results showed high levels of hexanal, nonanal, heptanal, octanal, and 1-octen-3-ol in fresh sea bass samples, with hexanal having an herbaceous taste and generally considered to be a representative substance of fishy taste [43]. Nonanal and octanal might be due to the oxidation of free fatty acids such as linoleic acid [44,45]. The detected 1-octen-3-ol has a mushroom aroma and influences the overall flavor and known to contribute to the characteristic mild, fresh, plant-like aromas of fresh fish [46]. It was found that the volatiles of the smoked products essentially consisted of the compounds detected in raw fish as well as other compounds produced during processing, as with the results of Guillén et al. [47]. The content of these compounds was notably lower in the salted sample (*p* < 0.05), while other compounds, such as n-octanal and 2-octanol, were not identified, and these phenomena may be related to the oxidation and enzymatic activity of the whole system [48]. However, the HS-GC-IMS results showed an increase in the concentration of aldehydes and alcohols, as with the results of Huang [20]. The specific reasons for the opposite results of the two methods need to be further investigated.

The volatile compounds of sea bass after drying were similar to those discovered before salting, despite their different concentrations. Nevertheless, (Z)-7-hexadecenal, trans-2-undecen-1-ol and palmitoleic acid were the new volatile compounds in dried samples. The content of hexanal was higher in fresh, salted, and dried samples, and it was relatively lower in smoked and heated samples, indicating that the components in the smoke flavors had a coordinated and balanced effect on the volatile compounds of fish. In the smoked sample, the contents of 2,3-pentanedione and 2-butanone decreased significantly (*p* < 0.05), which may be affected by microbial activity [49,50]. However, the smoked samples contained phenolics not found in fresh sea bass, such as guaiacol (2-ethylphenol) and eugenol. These phenolics are derived from the smoky flavor and have been identified as the most characteristic smoke-related components in smoked fish [51]. Notably, new volatile compounds were identified in the smoked samples because of further interactions of the components in the smoke flavors with proteins, peptides and free amino acids, which are characteristic compounds of the smoked flavor [52]. Benzaldehyde was present at low levels in unsmoked fish, but it was elevated after smoking, indicating that benzaldehyde was present in both unsmoked fish and the smoke flavors, similarly to the findings of Hedberg et al. [53]. Ketones are the products of the oxidation of unsaturated fatty acids and the Maillard reaction [54]. Hydroxyl acetone, 2,3-dimethylcyclopent-2-en-1-one, and 3-methyl-2-cyclopenten-1-one were more likely to be derived from the smoke flavors, where 3-methyl-2-cyclopentene-1-one has a typical scorched taste [55]. Saldaña et al. [56] found that the ketones from 2-cyclopenten-1-one were most likely derived from the Maillard reaction of cellulose pyrolysis during fumigation. At the same time, hexanal, 2-butanone and 2,3-pentanedione disappeared after “Smoke flavoring”, which may be due to the bad smell covered by the smoke flavors, or some chemical reactions occurred between the organic components in the “smoke flavoring”.

In addition, higher levels of undecane, dodecane, tridecane and heptadecane were detected in smoked fish flesh than in unsmoked fish flesh, which may be related to the lipid precursors in unsmoked fish [57,58]. The content of aldehydes (benzaldehyde), ketones (3-hydroxy-2-butanone) and alcohols (1-propanol) increased significantly with the processing of smoke-flavored sea bass, which was probably due to irreversible chemical reactions such as Strecker degradation or lipid oxidation. The precursors of butanal in fish flesh are mainly oleic (MUFA n-9) and linoleic acid (PUFA n-6) or its methyl esters [59]. Five saturated linear aldehydes, hexanal, heptanal, octanal, nonanal and valeraldehyde, had been identified as characteristic volatile compounds [60]. Although aliphatic aldehydes such as hexanal and glutaraldehyde have been found in wood smoke, most of them come from the oxidation of lipids in fish during the smoking process [58] and interact with proteins [61]. It has been found that heat treatment at higher temperatures can facilitate the production of aldehydes, which can degrade polyunsaturated fatty acids (linoleic acid, linolenic acid) more quickly, resulting in the production of hexanal, heptanal and other aldehydes. Simultaneously, fatty acids oxidation leads to an increase in the content of alcohols [62], whose odor threshold is usually higher than that of aldehydes, and their special flavor (floral, fruity) would promote the formation of fish aroma. The detected 1-pentanol had a significant effect on flavor [63]. When fish was processed, saturated ketones with specific flavor (fruity, cheesy) and diketones with caramel or other flavors may be associated with the degradation of amino acids and oxidation of polyunsaturated fatty acids [64]. The results showed that the contents of volatile compounds in smoke-flavored sea bass were significantly increased in dried and heated samples, which were more due to the significant increase of ketones (Figure 5). Ketones mainly came from the oxidation of polyunsaturated fatty acids or the degradation or oxidation of amino acids [64]. Saturated ketones had a unique fragrance, cheesy and fruity flavor, whereas diketones had a sweet, buttery, and caramel flavor [65]. It is worth noting that some compounds were also sensitive to temperature and easily decomposed or degraded in the presence of high temperature, which causes some volatile compounds to be lost or weakened in the heated sample. This finding agreed with Salum et al. [52] and Cantalejo et al. [66] that a slight aroma loss occurs during prolonged high-temperature processing. Consequently, it was important to demonstrate the changes in aroma precursors during processing [67].

## 5. Conclusions

This study aimed to elaborate the changes of volatile compounds in smoke-flavored sea bass in different production steps (raw, salted, dried, smoked and heated). A total of 63 volatiles were identified by HS-SPME-GC-MS; meanwhile, 26 volatiles were identified by HS-GC-IMS. The results suggested that 22 volatiles may be considered as potential key compounds. Among them, the salting process was beneficial in reducing the content of substances with fishy flavor (hexanal). The volatile compounds in the dried, smoked, and heated samples varied significantly. Further research into the processing conditions related to the transformation of distinctive volatile compounds may help to enhance the processing procedures as well as the quality and flavor of the smoke-flavored sea bass product.

## Figures and Tables

**Figure 1 foods-11-02614-f001:**
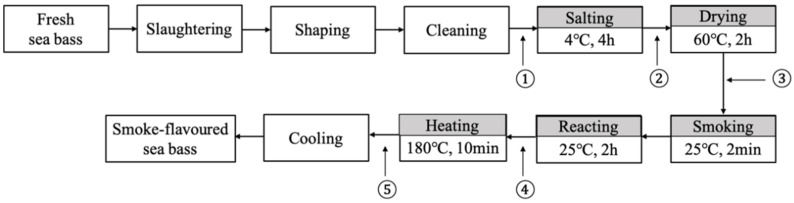
Schematic diagram of smoke-flavored sea bass processing line and five sampling points. Sampling point 1: fresh fish fillet; Sampling point 2: salting; Sampling point 3: drying; Sampling point 4: smoking; Sampling point 5: heating.

**Figure 2 foods-11-02614-f002:**
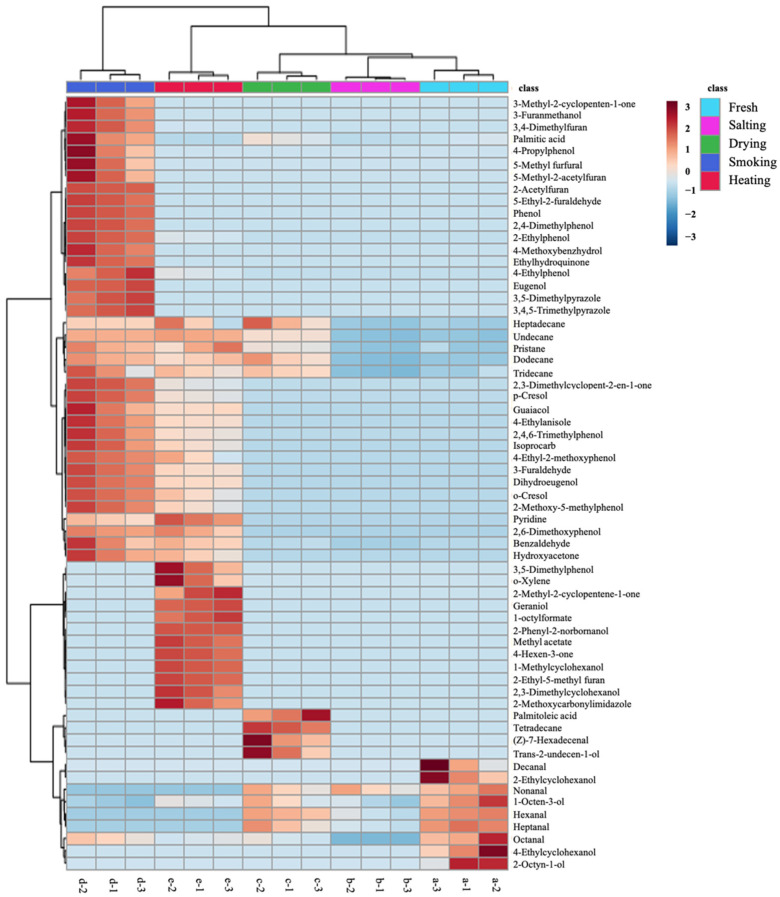
The heat map and clustering results of 63 volatile compounds in different production steps. Color shading from red to blue indicates abundances of compounds from high to low.

**Figure 3 foods-11-02614-f003:**
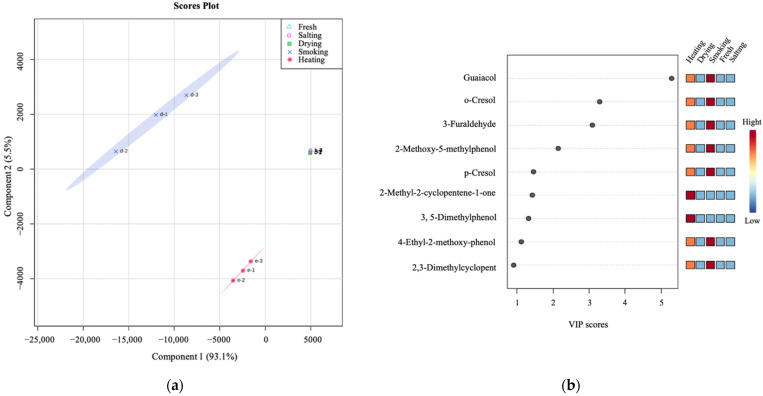
PLS-DA and the important flavors (VIP ≥ 1.0) for smoke-flavored sea bass at different production steps. (**a**) PLS-DA scores scatter plot. (**b**) Important volatiles (VIP ≥ 1.0) identified by PLS-DA. The colored boxes on the right indicate the relative concentrations of the corresponding volatiles at different stages.

**Figure 4 foods-11-02614-f004:**
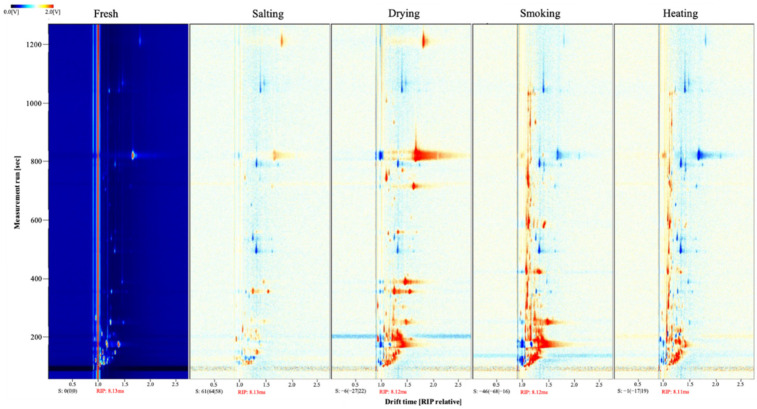
Two-dimensional (2D)-topographic subtraction plots based on the signal intensity of smoke-flavored sea bass with different production steps via HS-GC-IMS. The horizontal axis represents the ion migration time (ms), and the vertical axis represents the retention time (s).

**Figure 5 foods-11-02614-f005:**
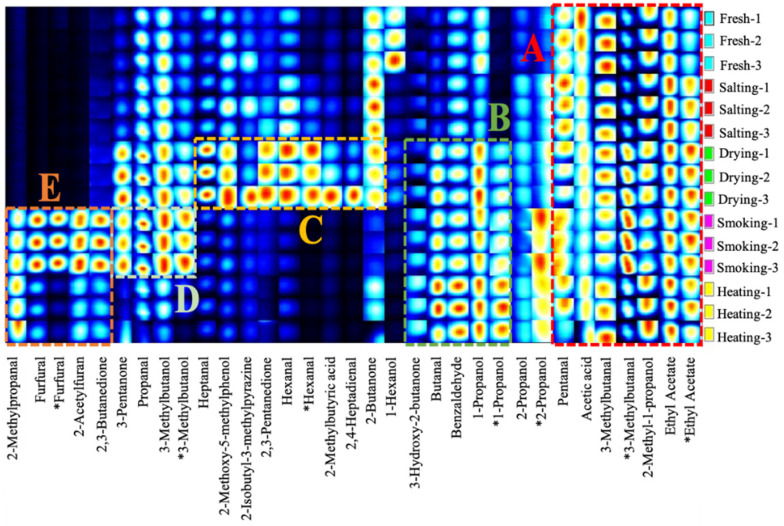
Gallery plot based on the signal intensity of smoke-flavored sea bass with different production steps via HS-GC-IMS. * Dimers formed in the IMS drift tube were represented by symbol “*”.

**Figure 6 foods-11-02614-f006:**
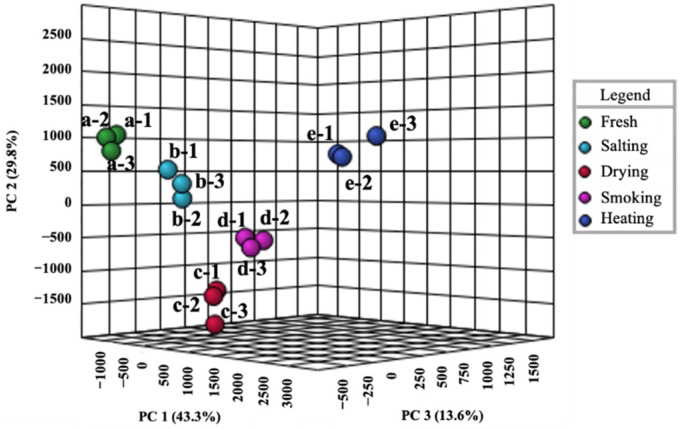
PCA based on the signal intensity of smoke-flavored sea bass with different production steps via HS-GC-IMS. a: fresh fish; b: salting; c: drying; d: smoking; e: heating.

**Figure 7 foods-11-02614-f007:**
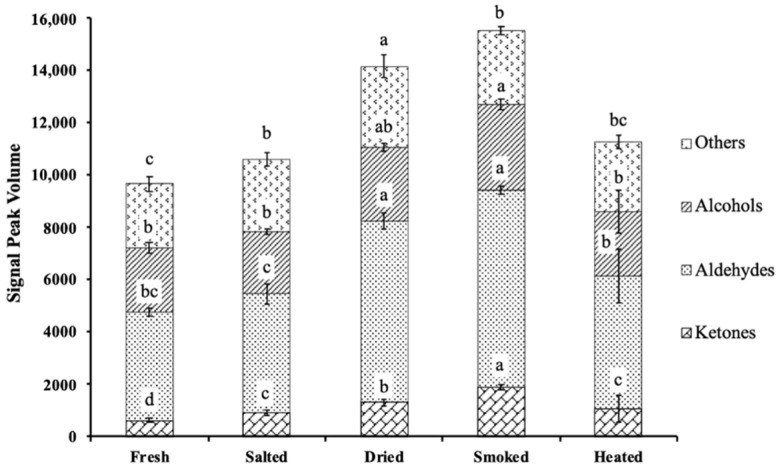
The signal peak volume distribution diagram of volatile compounds of smoke-flavored sea bass with different production steps via HS-GC-IMS. Values with different letters differ significantly (*p* < 0.05).

**Table 1 foods-11-02614-t001:** Experimental conditions for smoke-flavored sea bass analysis by HS-GC-IMS.

Gas Phase-Ion Mobility Spectrometry Unit	
Analysis time	30 min
Column type	MXT-5 (15 m × 0.53 mm)
Column temperature	60 °C
Carrier gas/drift gas	N2 (99.99%)
IMS temperature	45 °C
Automatic headspace sampling unit	
Injection volume	500 μL
Incubation time	20 min
Incubation temperature	40 °C
Syringe temperature	85 °C
Incubation speed	500 rpm

## Data Availability

The data presented in this study are available in the article.

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
