# Peer review of "Insight into the Characterization of Volatile Compounds in Smoke-Flavored Sea Bass (Lateolabrax maculatus) during Processing via HS-SPME-GC-MS and HS-GC-IMS"

_foods, 2022, doi:10.3390/foods11172614_

Round 1

Reviewer 1 Report

The objective of the research is clear. The analyzes are well structured, with a strong statistical component. The observations to the work are the following

1.- The presence of volatile compounds is associated with the processes, so it would have been convenient to include a process where only unsmoked fish was used to establish the compounds generated during this process. It is known that the addition of salt increases the volatile compounds during the headspace, and that the increase in temperature causes conformational changes in proteins that generate a modification in the interaction between volatile compounds and proteins. Therefore, it is important to establish which are the volatile compounds of the fish that undergo modifications during the process.

2.- Smoke is a mixture of volatile compounds, so it should also have been established which volatile compounds come from the smoke in the processed fish.

3.- It should be kept in mind that the extraction of volatile compounds with SPME includes 1 hour at 50°C (30 min of balance and 30 min of extraction). This additional hour generates volatile compounds resulting from biochemical reactions, associated with the analysis process and not with the processes involved.

4.-Volatile aldehyde compounds are generated during oxidation in meat products, and interact with proteins.

Author Response

Response to Reviewer 1 Comments

Point 1: The objective of the research is clear. The analyzes are well structured, with a strong statistical component. The observations to the work are the following.

Response 1: We are extremely grateful to the respected reviewer for the positive comments and helpful suggestions. We have carefully revised the manuscript according to your comments. We assume that the paper has been enhanced and we hope that the corrections made, along with the reply attached, will address the concerns raised. Below you will find our point-by-point responses to the comments/ questions addressed.

Point 2: The presence of volatile compounds is associated with the processes, so it would have been convenient to include a process where only unsmoked fish was used to establish the compounds generated during this process. It is known that the addition of salt increases the volatile compounds during the headspace, and that the increase in temperature causes conformational changes in proteins that generate a modification in the interaction between volatile compounds and proteins. Therefore, it is important to establish which are the volatile compounds of the fish that undergo modifications during the process.

Response 2: Thanks for your valuable comments. In our experimental design, volatile compounds in unsmoked fish have been analyzed, but their variation has not been analyzed in more depth, so we develop a more detailed discussion of some compounds in unsmoked fish in the discussion section of the revised manuscript to further demonstrate the presence of volatile compounds in relation to the process. As found in the study by Guille ́n et al. [1], the volatiles of the smoked fish basically comprise the compounds also detected in the raw fish together with others produced during the processing. In addition, the detected 1-octen-3-ol has a mushroom aroma and has an effect on the overall flavour and known to contribute to the characteristic mild, fresh, plant-like aromas of fresh fish [2]. Benzaldehyde and derived compounds were also found in both unsmoked fish and wood smoke [3], but only in a small proportion from unsmoked fish.

Point 3: Smoke is a mixture of volatile compounds, so it should also have been established which volatile compounds come from the smoke in the processed fish.

Response 3: Thanks for your valuable suggestion. Additions have been made in the revised draft. As you said, smoke and smoke flavorings contain a large number of components, derived from cellulose, hemicellulose and lignin pyrolysis, which have different functional groups, such as ketones, aldehydes, acids, ethers, hydrocarbons, carbohydrate derivatives, nitrogen derivatives, and phenol, guaiacol, syringol, and pyrocatechol derivatives [1]. The typical smoke flavours result from a number of chemicals found in the smoke, but is mostly attributed to the phenols. Phenolic compounds, which are mainly produced by pyrolysis of lignin, are important for preservation and flavour properties of smoked products. The content of phenolic compounds in these products depends on the nature of wood. Se ́rot et al. [4] found that phenolic derivatives such as guaiacol (2-methoxyphenol) and eugenol (2,6-dimethoxyphenol) were identified as the most characteristic smoke-related components in smoked fish. However, even if sometimes Maillard and Strecker products have been recovered in fish flesh, several studies have permitted to conclude that in general, cyclic compounds such as furfural, benzaldehyde, and all the derived molecules come from the wood smoke [1, 5].

It is important to note that aliphatic aldehydes are derived from lipid oxidation in fish. Even if small quantities of aliphatic aldehydes like butanal, pentanal, hexanal have been recovered in wood smoke in weak quantities [3], the most part of aliphatic volatile aldehydes in smoked fish comes from the fish flesh lipid oxidation under smoking process conditions [6]. In addition, in the smoking process the smoke components are adsorbed by the fish surface, and they react or establish interactions more or less strongly with fish components depending on the magnitude of the intermolecular forces existing between components of both systems. As consequence of this, the headspace of the smoked fish only will contain those smoke components, adsorbed on the fish surface, which have not reacted or interacted very strongly with fish components. That is to say the presence or absence of smoke components in the headspace of the smoked fish is governed by the interactions established between components of both systems [1].

Point 4: It should be kept in mind that the extraction of volatile compounds with SPME includes 1 hour at 50°C (30 min of balance and 30 min of extraction). This additional hour generates volatile compounds resulting from biochemical reactions, associated with the analysis process and not with the processes involved.

Response 4: Thanks for your clarification, we agree with you and will pay extra attention in our future experiments. In addition, there are many factors that affect SPME fiber performance, such as the choice of stationary phase and extraction conditions [7]. Solid phase extraction is a process that utilize a small amount of solvents, stripped out volatile compounds and recovered many compounds qualitatively [8]. Dynamic head-space is widely used in studies for the estimation flavor compounds in food since the 1980s. SPME is widely used as an alternative for extracting the volatile compounds of cooked meat. However, for now, more researchers using SPME to extract volatile compounds really do not take into account that some of the volatile compounds are produced by biochemical reactions that are relevant to the analytical process, so you gave good advice that will help us to complete volatile compounds with higher quality in the future.

Point 5: Volatile aldehyde compounds are generated during oxidation in meat products, and interact with proteins.

Response 5: Thanks for the clarification. It has been added in the revised manuscript now.

References

  1. Guillén, M. D.; Errecalde, M. C.; Salmerón, J.; Casas, C., Headspace volatile components of smoked swordfish (Xiphias gladius) and cod (Gadus morhua) detected by means of solid phase microextraction and gas chromatography–mass spectrometry. Food Chemistry. 2006, 94, (1), 151-156. https://doi.org/https://doi.org/10.1016/j.foodchem.2005.01.014.
  2. JOSEPHSON, D. B.; LINDSAY, R. C., c4-Heptenal: An Influential Volatile Compound in Boiled Potato Flavor. Journal of Food Science. 1987, 52, (2), 328-331. https://doi.org/https://doi.org/10.1111/j.1365-2621.1987.tb06605.x.
  3. Hedberg, E.; Kristensson, A.; Ohlsson, M.; Johansson, C.; Johansson, P.-Å.; Swietlicki, E.; Vesely, V.; Wideqvist, U.; Westerholm, R., Chemical and physical characterization of emissions from birch wood combustion in a wood stove. Atmospheric Environment. 2002, 36, (30), 4823-4837. https://doi.org/https://doi.org/10.1016/S1352-2310(02)00417-X.
  4. Sérot, T.; Baron, R.; Knockaert, C.; Vallet, J. L., Effect of smoking processes on the contents of 10 major phenolic compounds in smoked fillets of herring (Cuplea harengus). Food Chemistry. 2004, 85, (1), 111-120. https://doi.org/https://doi.org/10.1016/j.foodchem.2003.06.011.
  5. Varlet, V.; Serot, T.; Knockaert, C.; Cornet, J.; Cardinal, M.; Monteau, F.; Le Bizec, B.; Prost, C., Organoleptic characterization and PAH content of salmon (Salmo salar) fillets smoked according to four industrial smoking techniques. Journal of the Science of Food and Agriculture. 2007, 87, (5), 847-854. https://doi.org/https://doi.org/10.1002/jsfa.2786.
  6. Varlet, V.; Prost, C.; Serot, T., Volatile aldehydes in smoked fish: Analysis methods, occurence and mechanisms of formation. Food Chemistry. 2007, 105, 1536-1556. https://doi.org/10.1016/j.foodchem.2007.03.041.
  7. Lorenzo, J. M., Influence of the type of fiber coating and extraction time on foal dry-cured loin volatile compounds extracted by solid-phase microextraction (SPME). Meat Science. 2014, 96, (1), 179-186. https://doi.org/https://doi.org/10.1016/j.meatsci.2013.06.017.
  8. Madruga, M. S.; Stephen Elmore, J.; Dodson, A. T.; Mottram, D. S., Volatile flavour profile of goat meat extracted by three widely used techniques. Food Chemistry. 2009, 115, (3), 1081-1087. https://doi.org/https://doi.org/10.1016/j.foodchem.2008.12.065.

Reviewer 2 Report

The topic of this manuscript is interesting and timely. Volatile compounds in smoke-flavoured sea bass during processing via HS-SPME-GC- MS and HS-GC-IMS were examined.

A lot of considerations about different aspects of the manuscript should be done.I report many ideas, suggestions, constructive criticism. Unfortunately, the manuscript has several contradictions, many sentences are repeated. In some cases, the reader may be confused in reading and understanding some concepts.

It’s not understandable why the authors made some technical choices; I refer, for example, to the cooking of the samples after smoke-flvouring of fish samples. The heat treatment is an unusual practice after smoking. The authors did not give a justification for this choice.

The authors paid little attention to the origin of the volatile compounds, although they show some figures referred to an advanced statistical approach.

There is a lack of references.

English language must be improved. The revision of this manuscript is detailed below.

Abstract:  The aim of this study must be related to the aim reported in lines 68-69.

Title and line 11: it would be better to write the scientific name of sea bass

Line 14: n.85 compounds were identified; probably volatiles were 86.

The authors write “…..and 25 distinctive flavourings with significant processing contributions were screened”; this sentence should be changed, and the whole context will become clearer.

Line 32: the hazards should be considered as “potential”; please, change the sentence

Line 33: the “smoke -flavouring” process cannot be considered as “recently” applied; the authors report the reference [1] dating back to 2008. In the context of this publication the authors stated that “Smoke flavorings have evolved as a successful alternative to traditional smoking

Line 34: “According to the survey [3]”….should be written as “ According to the survey of Portella et al.[3],.

Furthermore, it would be appropriate to report more up-to-date data than Portella et al. (2011), if available.

Line: 44: other factors can influence the flavor of smoked fish, first the quality of the raw fish should be considered. The authors should underline this aspect

Line 47: smoking temperature is a factor not to be neglected. Mot constantly the process of modern smoking results “in a lack of flavor in smoked product”. The authors should support this assertion by one or more references.

Line 53: [9- 10 - 11] is more correct

Line 61: [18-19-20-21] is more correct

From line 58 to line 66: I agree with the authors’ opinion (as supported by references; it’s also true that other researchers have in the past applied more sensitive techniques in combination with statistical data analysis. This was also the case in the periods prior to 2019 or 2020, as I have seen in the references. Authors could also include some references from before 2019, if possible.

From line 67 to 72: the aim of the paper is not clear because some sentences refer to the content of “Materials and Methods” and “Results”. The sentences need to be rephrased

Line 77: it would be advisable to specify how the samples were transported to the laboratory

Line 78: “decapitated and cleaned”; the authors should write that “the fresh sea bass were slaughtered, headed, degutted and washed”

Lines 88-89: the sentence “Samples of sea bass were collected for analysis after five of the key stages in smoke flavored sea bass production (Figure 1)” must be written after the processing line description.

In Figure 1 the schematic processing line is reported; the final treatment is the sample cooling. In line 88 the authors state that “samples of sea bass were collected for analysis after five of the key stages….”. I ask why the cooled samples were not collected?

A further consideration concerns the smoke-flavouring of samples: the authors partially followed the procedure described by Çakir and Ayvaz (2020) for anchovy samples. After salting, the samples treatment is different to that applied by Çakir and Ayvaz (2020), except for heating at 180 °C for 10 min.

The procedure used by the authors raises some doubts: 1) anchovies weigh is less than the tilapia samples (average weight of the fish was 534.5 g, as reported in the study); consequently, the application time (10 min) on fish flesh is different in tilapia than in anchovy; 2) after salting the samples are treated using a procedure that is different to that applied by Çakir and Ayvaz but no reference was reported in this section ; 3) why did the authors cook the samples? I ask this because the liquid smoke-flavouring is usually not followed by cooking. Çakir and Ayvaz (2020), cited by the authors, cooked their samples to monitor the color changes of fillets at various stages of two smoking processes and after samples cooking.

The following book chapter could be useful: Dincer, Ibrahim; Midilli, Adnan; Kucuk, Haydar (2014). Progress in Sustainable Energy Technologies Vol II.  Environmental Friendly Food Smoking Technologies., 10.1007/978-3-319-07977-6(Chapter 37), 557–576. doi:10.1007/978-3-319-07977-6_37

Table 1: “Title 2” - the author should find a caption for the relevant column

Line 99: the fresh fish fillets were not included in the paragraph 2.2; in general, the authors do not write about the difference between fresh and treated samples

Line 99 (HS-SPME-HS-SPME-GC-MS analysis) and line 123 (HS-GC-IMS analysis): what was the basis for the analytical techniques? I referr to the weight of the sample, the temperatures and the time taken, etc.

Line 115: it is advisable to specify parameters of instrumental analysis; in my opinion the reader must clearly understand the analytical approach.

Lines 133: “Statistical analysis” does not refer to the correlation between the analytical methods (two) used.

But, in lines 68-69 the authors write “The diversity of volatile substances in smoke-flavored sea bass was obtained by using a combination of HS-SPME-GC-MS and HS-GC-IMS and the correlation between the two methods was examined”.

Figures 2 and 3 should have a better definition in the figures caption. Furthermore, it would be better to give an order to the samples based on different treatment (i.e. fresh, salting, etc.)

Figure 6 needs an improvement in the definition of figure caption

A lower number of figures would have sufficed to show the trend of volatiles during the different production steps. The authors give an extensive description of the compounds with different production steps.

Figure 6 - PCA analysis: the reader cannot distinguish the denomination of compounds reported in the center of the image. A three-dimensional PCA would have allowed to obtain a better cumulative variance contribution rate

Line 187: the authors should explain the acronym VIP

Line 210: the authors write “The smoke-flavoured sea bass has a total of 26 volatile compounds that have been tentatively identified (Table S2)”; with 7 volatiles (in both monomer and dimer forms), the compounds are 33, but in Figure 5 they are 34.

Line 288: the authors write “The combination of flavour analysis techniques…”; in lines 293- 294 you can read “…were detected and analyzed by two methods”.  The sentences need to be clearer: is it a combination of two techniques or two distinct methods were used?

Line 294: please, check the compounds number. N.63 compounds were identified by the first method and 33 by the second one, with a total of 86 compounds.

Line 306-307: “the combination of these two methods provides a more comprehensive aroma characterization”. Although this assertion may be agreeable, in my opinion these combined techniques cannot be proposed for a practical application (i.e. for the industry): they can time consuming, expensive and require a great deal of precision and, sometimes, the repetition of analyses.

Line 327:”…. microorganisms’ activity”. The statement needs a reference.

Line 336: “….has a typical scorched taste…” .The statement needs a reference.

Line 339: the end of the sentence needs a reference.

Line 356 to 359:”….. not be ignored”. The sentences need a reference

From line 361 to 366: one o more references are needed

From line 368 to 371: the sentence repeats concepts already expressed earlier by the authors

Line 369: I wonder what is one of the advantages of HS-GC-IMS because it is less sensitive and detects fewer compounds than HS-SPME-GC-MS

Conclusions: some sentences are repeated. This section should be shorter.

Reference section: must be checked carefully. There are numerous flaws 

Author Response

Response to Reviewer 2 Comments

Point 1: The topic of this manuscript is interesting and timely. Volatile compounds in smoke-flavoured sea bass during processing via HS-SPME-GC- MS and HS-GC-IMS were examined.

A lot of considerations about different aspects of the manuscript should be done. I report many ideas, suggestions, constructive criticism. Unfortunately, the manuscript has several contradictions, many sentences are repeated. In some cases, the reader may be confused in reading and understanding some concepts.

It’s not understandable why the authors made some technical choices; I refer, for example, to the cooking of the samples after smoke-flvouring of fish samples. The heat treatment is an unusual practice after smoking. The authors did not give a justification for this choice.

The authors paid little attention to the origin of the volatile compounds, although they show some figures referred to an advanced statistical approach.

There is a lack of references.

English language must be improved. The revision of this manuscript is detailed below.

Response 1: We are extremely grateful to the respected reviewer for the positive comments and helpful suggestions. These suggestions are indeed very useful, valuable and helpful for the revision and improvement of our paper. We observed the comments carefully and made the correction based on the comments of the editor and the reviewer. The major corrections in the paper are marked with a red color, and the responses to the comments of the reviewer are as follows.

Abstract:

Point 2: The aim of this study must be related to the aim reported in lines 68-69.

Response 2: Thank you for your careful observation. The aim has been modified and rephrased in the revised manuscript now.

Point 3: Title and line 11: it would be better to write the scientific name of sea bass.

Response 3: Thanks for your valuable comments. We have added the scientific name of sea bass

Point 4: Line 14: n.85 compounds were identified; probably volatiles were 86.

Response 4: Thanks for your careful observations. We have reconfirmed that a total of 85 compounds have been identified.

Point 5: The authors write “…..and 25 distinctive flavourings with significant processing contributions were screened”; this sentence should be changed, and the whole context will become clearer.

Response 5: Thanks for your valuable comments. This section has been rephrased as “and 25 compounds that may be considered as potential key compounds were screened by principal component analysis (PCA) and partial least squares discriminant analysis (PLS-DA).” in our revised manuscript. Please check it.

Point 6: Line 32: the hazards should be considered as “potential”; please, change the sentence.

Response 6: Thanks for your valuable suggestion. It has been modified in the revised manuscript now.

Point 7: Line 33: the “smoke -flavouring” process cannot be considered as “recently” applied; the authors report the reference dating back to 2008. In the context of this publication the authors stated that “Smoke flavorings have evolved as a successful alternative to traditional smoking”.

Response 7: Thanks for your valuable suggestion. It has been modified in the revised manuscript now.

Point 8: Line 34: “According to the survey [3]”….should be written as “ According to the survey of Portella et al.[3],. Furthermore, it would be appropriate to report more up-to-date data than Portella et al. (2011), if available.

Response 8: Thanks for your careful observations. The corresponding introduction part have been rewritten, and some recent studies have been added now.

Point 9: Line: 44: other factors can influence the flavor of smoked fish, first the quality of the raw fish should be considered. The authors should underline this aspect.

Response 9: Thanks for your valuable comments. The statement that “The quality of smoke-flavoured sea bass and the formation of its flavor are related to the characteristics of raw fish and production operations, such as the variability of raw materials (fat content), brine concentration, and smoking techniques.” have been added in our revised manuscript and highlighted in a red color. Please check it.

Point 10: Line 47: smoking temperature is a factor not to be neglected. Mot constantly the process of modern smoking results “in a lack of flavor in smoked product”. The authors should support this assertion by one or more references.

Response 10: Thanks for your careful observation. In a previous study, Cardinal et al. [1] have shown that fillets smoked with liquid smoke were characterised by “salmon-like” odour one time and a half weaker than the odour of fillets smoked by smouldering process (intensity of 2.3 for liquid smoke and 4.3 for smouldering). This conclusion was also confirmed by Varlet et al. [2]. Similar works have been cited in our revised manuscript. Please check it.

Point 11: Line 53: [9- 10 - 11] is more correct.

Response 11: It has been modified now. Thanks.

Point 12: Line 61: [18-19-20-21] is more correct.

Response 12: It has been modified now. Thanks.

Point 13: From line 58 to line 66: I agree with the authors’ opinion (as supported by references; it’s also true that other researchers have in the past applied more sensitive techniques in combination with statistical data analysis. This was also the case in the periods prior to 2019 or 2020, as I have seen in the references. Authors could also include some references from before 2019, if possible.

Response 13: Thanks for your careful observations. The corresponding introduction part have been rewritten, and some references from before 2019 have been added now.

Point 14: From line 67 to 72: the aim of the paper is not clear because some sentences refer to the content of “Materials and Methods” and “Results”. The sentences need to be rephrased.

Response 14: Thanks for your valuable comments. The aim has been rephrased as “Hence, this work investigated the changes of volatile compounds in smoked sea bass at different stages of processing. The diversity of volatile substances in smoke-flavoured sea bass was obtained by using a combination of HS-SPME-GC-MS and HS-GC-IMS, and the chemical fingerprints of smoke-flavoured sea bass was also obtained. Furthermore, principal component analysis (PCA) and partial least squares discriminant analysis (PLS-DA) were performed to elucidate the correlation between volatile compounds and different processing stages. Overall, these studies may help to further improve the flavor quality of smoke-flavoured fish.” in our revised manuscript. Please check it.

Point 15: Line 77: it would be advisable to specify how the samples were transported to the laboratory.

Response 15: Thanks for your helpful remarks. It has been added now in the revised manuscript now.

Point 16: Line 78: “decapitated and cleaned”; the authors should write that “the fresh sea bass were slaughtered, headed, degutted and washed”.

Response 16: Thanks for your correction. It has been revised as “the fresh sea bass were slaughtered, headed, degutted and washed” in our revised manuscript.

Point 17: Lines 88-89: the sentence “Samples of sea bass were collected for analysis after five of the key stages in smoke flavored sea bass production (Figure 1)” must be written after the processing line description.

Response 17: Thanks for your valuable comments. It has been modified now in the revised manuscript.

Point 18: In Figure 1 the schematic processing line is reported; the final treatment is the sample cooling. In line 88 the authors state that “samples of sea bass were collected for analysis after five of the key stages….”. I ask why the cooled samples were not collected?

Response 18: Thanks for your careful observations. In fact, the "Cooling" in the processing line of Figure 1 is the natural cooling of the samples to room temperature and is not a critical processing point. Although we used five key stages for the sea bass samples, all samples were placed in the same environment for natural resting before analysis. By this I mean that the actual temperature of all samples before analysis was close to room temperature. Just as Yao et al. [3] analyzed the flavor formation during production of Dezhou braised chicken, although there was a "Cooling" process in the production line, they did not sample after this process, but chose seven other key processing points. Similarly, Hu et al. [4] analyzed the flavor characteristics of dried shrimp at different processing stages, they did not sample at the process of cooling, as did Chen et al. [5] and Cui et al. [6]. Therefore, we selected only five key processing points for sampling, and we hope to meet with your approval on the response. Thanks for your valuable comments again.

Point 19: A further consideration concerns the smoke-flavouring of samples: the authors partially followed the procedure described by Çakir and Ayvaz (2020) for anchovy samples. After salting, the samples treatment is different to that applied by Çakir and Ayvaz (2020), except for heating at 180 °C for 10 min.

Response 19: Thanks for your careful observation. We clarified the reference procedure for the smoke-flavouring method and our research team explored and optimized the entire process parameters prior to this experiment. For details, please check the manuscript. The method of Çakir et al. [7] consisted of immersing the cleaned fillets in a 10% brine solution (1:1, w/v) at 4 ± 2°C for 4 hours. They were removed from the brining solution and dried at room temperature for 10 min. They were then immersed in liquid smoke condensate for 1, 2 and 3 min depending on the prespecified group. Then, they were cooked in a fan oven at 180°C for 10 minutes. But we found that sea bass fillets smoked after drying at room temperature for 10 minutes were much less effective than drying in a hot air drying oven, which may be similar to the hot or cold smoking process [8], where light drying is necessary before smoking. Because a dry surface absorbs the soot better, the surface of smoked fish forms the desired color,so we refer to the drying process. Liquid smoking was “immersed in a solution of smoke flavouring in water (1:5, v/v) for 2 min with a fish-to-flavouring solution proportion of 1:15 (w/v)”. Although the concentration of the smoke flavours may be inconsistent, the appropriate concentration was chosen to achieve the desired organoleptic effect. Afterwards, smoked fillets were kept at room temperature for 2 h to facilitate the interaction between smoke components and fish flesh [9]. We hope to meet with your approval on the revision.

Point 20: The procedure used by the authors raises some doubts: 1) anchovies weigh is less than the tilapia samples (average weight of the fish was 534.5 g, as reported in the study); consequently, the application time (10 min) on fish flesh is different in tilapia than in anchovy; 2) after salting the samples are treated using a procedure that is different to that applied by Çakir and Ayvaz but no reference was reported in this section ; 3) why did the authors cook the samples? I ask this because the liquid smoke-flavouring is usually not followed by cooking. Çakir and Ayvaz (2020), cited by the authors, cooked their samples to monitor the color changes of fillets at various stages of two smoking processes and after samples cooking. The following book chapter could be useful: Dincer, Ibrahim; Midilli, Adnan; Kucuk, Haydar (2014). Progress in Sustainable Energy Technologies Vol II. Environmental Friendly Food Smoking Technologies., 10.1007/978-3-319-07977-6(Chapter 37), 557–576. doi:10.1007/978-3-319-07977-6_37

Response 20: Thanks for your careful observation. We hope to meet with your approval on the revision. 1)Although the average weight of the fish was 534.5 g (± 32.8 g), the average weight of the fillets after pre-treatment was 110.5 g (± 8.5 g), which has been added in the methods section of the revised manuscript. Even so the weight was still much higher than that of anchovies, but the smoked flavored sea bass processed under this condition was the most acceptable after our pre-treatment iterations; 2) We clarified the reference procedure for the smoke-flavouring method. Additions have now been made in the revised draft; 3) Thanks for your precious and valuable explanation. I agree with you that " the liquid smoke-flavouring is usually not followed by cooking ". However, this operation is more suitable for fish that can be eaten directly, such as salmon, etc. To prepare liquid smoked bonito, Dien et al. [10] soaked fresh fish fillets in a 1% liquid smoke solution for 5 min, then drained and heated in an oven at 150°C for 1 h. Çakir et al. [7] also performed the cooking process. And, prior to this experiment, our research group explored and optimized the whole process parameters and the cooked samples were acceptable to everyone. We hope to meet with your approval on the revision.

Point 21: Table 1: “Title 2” - the author should find a caption for the relevant column.

Response 21: Thanks for more clarification. The "Title 2" in the table was entered incorrectly, thus we have already deleted it in our revised manuscript.

Point 22: Line 99: the fresh fish fillets were not included in the paragraph 2.2; in general, the authors do not write about the difference between fresh and treated samples.

Response 22: Thanks for your careful observation. We have added some details to paragraph 2.2 by emphasizing that the samples are from different sampling points, including fresh samples. Please check it.

Point 23: Line 99 (HS-SPME-HS-SPME-GC-MS analysis) and line 123 (HS-GC-IMS analysis): what was the basis for the analytical techniques? I referr to the weight of the sample, the temperatures and the time taken, etc.

Response 23: Thanks for your precious and valuable explanation. Additions have now been made in the revised draft. In HS-SPME-GC-MS analysis, " Four grams of samples from different sampling points were transferred into head-space vial (20 ml)" and " The mixture was then balanced at 50 °C, for 30 min " contain the weight of the sample, the temperature and the time taken. In HS-GC-IMS analysis, “Two grams of samples from different sampling points were transferred into head-space vial (20 ml)” and “Table 1 Incubation time 20, Incubation temperature 40 °C” contain the weight of the sample, the temperature and the time taken. Please check it.

Point 24: Line 115: it is advisable to specify parameters of instrumental analysis; in my opinion the reader must clearly understand the analytical approach.

Response 24: Thanks for your valuable comments. The analytical approach had been clearly understood in our revised manuscript.

Point 25: Lines 133: “Statistical analysis” does not refer to the correlation between the analytical methods (two) used. But, in lines 68-69 the authors write “The diversity of volatile substances in smoke-flavored sea bass was obtained by using a combination of HS-SPME-GC-MS and HS-GC-IMS and the correlation between the two methods was examined”.

Response 25: Thanks for your careful observation. The aim section and statistical analysis have now been modified and rephrased in the revised manuscript now. Please check it.

Point 26: Figures 2 and 3 should have a better definition in the figures caption. Furthermore, it would be better to give an order to the samples based on different treatment (i.e. fresh, salting, etc.).

Response 26: Thanks for your careful observation. We have added some details in the figure and table captions as your suggestion with red color. However, Heatmap analysis were performed in MetaboAnalyst 5.0 (https://www.metaboanalyst.ca), and we may not be able to customize the sorting by processing order, but it is close enough. Please check.

Point 27: Figure 6 needs an improvement in the definition of figure caption

A lower number of figures would have sufficed to show the trend of volatiles during the different production steps. The authors give an extensive description of the compounds with different production steps.

Response 27: Thanks for your valuable comments. We have added some details in the figure as your suggestion with red color.

Point 28: Figure 6 - PCA analysis: the reader cannot distinguish the denomination of compounds reported in the center of the image. A three-dimensional PCA would have allowed to obtain a better cumulative variance contribution rate.

Response 28: It has been improved now. Thanks.

Point 29: Line 187: the authors should explain the acronym VIP.

Response 29: Thanks for your careful observation. It has been modified now in the revised manuscript.

Point 30: Line 210: the authors write “The smoke-flavoured sea bass has a total of 26 volatile compounds that have been tentatively identified (Table S2)”; with 7 volatiles (in both monomer and dimer forms), the compounds are 33, but in Figure 5 they are 34.

Response 30: Thanks for your careful observation. We previously had some problems with the analysis of volatile substances in Figure 5, to be precise, there should be 33 volatile substances, which have now been revised in the revised manuscript.

Point 31: Line 288: the authors write “The combination of flavour analysis techniques…”; in lines 293- 294 you can read “…were detected and analyzed by two methods”.  The sentences need to be clearer: is it a combination of two techniques or two distinct methods were used?

Response 31: Thanks for your careful observation. It has been modified now in the revised manuscript. It should be " The combination of flavour analysis techniques…".

Point 32: Line 294: please, check the compounds number. N.63 compounds were identified by the first method and 33 by the second one, with a total of 86 compounds.

Response 32: Thanks for your careful observation. It has been modified now in the revised manuscript. We checked the compound numbers. N.63 compounds were identified by the first method and 33 by the second one, but 7 of the 33 compounds identified by the second method were in the form of dimers, so 26 compounds were actually detected using the second method, and then the 4 compounds detected by both methods together were removed to end up with a total of 85 compounds. In addition, another representation of the dimer (* Dimers formed in the IMS drift tube were represented by symbol “*”) in Figure 5 and in the supplementary material (Table S2) was used for a better understanding by the reader.

Point 33: Line 306-307: “the combination of these two methods provides a more comprehensive aroma characterization”. Although this assertion may be agreeable, in my opinion these combined techniques cannot be proposed for a practical application (i.e. for the industry): they can time consuming, expensive and require a great deal of precision and, sometimes, the repetition of analyses.

Response 33: Thanks for your valuable suggestion. We have considered these issues before, but HS-SPME-GC-MS is a combined technique of extraction with separation detection to qualitatively and quantitatively detect VOCs in foods. It has been widely applied in the field of food analysis such as traceability analysis and identification of varieties [11] and VOCs in fish fillets during cold storage [12]. However, it requires tedious pre-processing and long detection time, which limits the efficacy of GC-MS [13]. GC-IMS combined the high separation ability of GC and the fast response of ion mobility spectrometry (IMS), which had the advantages of low detection limit, no requirement for the pretreatment of samples, secondary separation, and displaying results intuitively [14]. Chen et al. [15] analyzed the effects of four types of thermal processing methods on the aroma profiles of acidity regulator-treated tilapia muscles and similarly concluded that the combination of HS-SPME-GC-MS and HS-GC-IMS could more comprehensive aroma profile than either of these two methods. However, the shortcoming of HS-GC-IMS is that all VOCs cannot be identified due to the imperfection of NIST and IMS databases. With the development of NIST and IMS databases, the research work will be greatly improved, and better information may be obtained in practical applications (i.e. for the industry) with high efficiency and low cost.

Point 34: Line 327:”…. microorganisms’ activity”. The statement needs a reference.

Response 34: It has been improved now. Thanks.

Point 35: Line 336: “….has a typical scorched taste…” .The statement needs a reference.

Response 35: It has been improved now. Thanks.

Point 36: Line 339: the end of the sentence needs a reference.

Response 36: Thanks for your careful observation. Varlet et al. [2] found that liquid smoke can mask the fishy odor of smoked salmon meat itself. The same remark could be formulated to explain the weaker perception of ‘butter’ odor in products smoked with liquid smoke. Therefore, we speculate that the reduction or disappearance of some of the volatiles is due to the action of the smoking solution. Similar works have been cited in our revised manuscript. Please check it.

Point 37: Line 356 to 359:”….. not be ignored”. The sentences need a reference.

Response 37: Thanks for your valuable suggestions. Lines 356 to 359 were duplicated above and have now been removed and new details have been added. Please check.

Point 38: From line 361 to 366: one o more references are needed.

Response 38: It has been improved now. Thanks.

Point 39: From line 368 to 371: the sentence repeats concepts already expressed earlier by the authors.

Response 39: Thanks for your valuable suggestion. That sentence has been removed in the revised manuscript now.

Point 40: Line 369: I wonder what is one of the advantages of HS-GC-IMS because it is less sensitive and detects fewer compounds than HS-SPME-GC-MS.

Response 40: Thanks for your consideration. GC-IMS combined the high separation ability of GC and the fast response of ion mobility spectrometry (IMS), which had the advantages of low detection limit, no requirement for the pretreatment of samples, secondary separation, and displaying results intuitively [14]. It should be noted that IMS is most suitable for trace gas analysis. The analytical device has proven attractive due to its lower detection limits, extraordinary sensitivity and its ease of use. These advantages have improved the application ranges of IMS instrument. GC-IMS is an inexpensive and powerful analytical technique for the detection of flavor compounds at ambient pressures and temperatures. The advantages of HS-GC-IMS were described in the introductory section of the revised manuscript. Currently only the NIST and IMS databases are not well developed, so this results in fewer compounds being detected.

Point 41: Conclusions: some sentences are repeated. This section should be shorter.

Response 41: Thanks for your consideration. The conclusion section has been modified and rephrased in the revised manuscript now.

Point 42: Reference section: must be checked carefully. There are numerous flaws.

Response 42: Thanks for your careful observation. We double-checked the references section and made changes.

References

  1. Mireille, C.; Berdagué, J. L.; Dinel, V.; Knockaert, C.; Vallet, J. L., Effect of various smoking techniques on the nature of volatile compounds and on the sensory characteristics of salmon meat. Sciences des Aliments. 1997, 17, 679-696.
  2. Varlet, V.; Serot, T.; Knockaert, C.; Cornet, J.; Cardinal, M.; Monteau, F.; Le Bizec, B.; Prost, C., Organoleptic characterization and PAH content of salmon (Salmo salar) fillets smoked according to four industrial smoking techniques. Journal of the Science of Food and Agriculture. 2007, 87, (5), 847-854. https://doi.org/https://doi.org/10.1002/jsfa.2786.
  3. Yao, W.; Cai, Y.; Liu, D.; Chen, Y.; Li, J.; Zhang, M.; Chen, N.; Zhang, H., Analysis of flavor formation during production of Dezhou braised chicken using headspace-gas chromatography-ion mobility spec-trometry (HS-GC-IMS). Food Chemistry. 2022, 370, 130989. https://doi.org/https://doi.org/10.1016/j.foodchem.2021.130989.
  4. Hu, M. Y.; Wang, S. Y.; Liu, Q.; Cao, R.; Xue, Y., Flavor profile of dried shrimp at different processing stages. Lwt-Food Sci Technol. 2021, 146, 111403. https://doi.org/https://doi.org/10.1016/j.lwt.2021.111403.
  5. Chen, J. H.; Tao, L. N.; Zhang, T.; Zhang, J. J.; Wu, T. T.; Luan, D. L.; Ni, L.; Wang, X. C.; Zhong, J., Effect of four types of thermal processing methods on the aroma profiles of acidity regulator-treated tilapia muscles using E-nose, HS-SPME-GC-MS, and HS-GC-IMS. Lwt-Food Sci Technol. 2021, 147, 111585. https://doi.org/https://doi.org/10.1016/j.lwt.2021.111585.
  6. Cui, Z.; Yan, H.; Manoli, T.; Mo, H.; Li, H.; Zhang, H., Changes in the volatile components of squid (illex argentinus) for different cooking methods via headspace–gas chromatography–ion mobility spectrometry. Food Science & Nutrition. 2020, 8, (10), 5748-5762. https://doi.org/https://doi.org/10.1002/fsn3.1877.
  7. Çakir, F.; Ayvaz, Z., Investigation of the Effect of Different Immersion Times of Anchovy Fillets in Liquid Smoke Flavoring on Color by Image Analysis. Journal of Aquatic Food Product Technology. 2020, 29, (9), 865-870. https://doi.org/10.1080/10498850.2020.1813857.
  8. Belichovska, K.; Belichovska, D.; Pejkovski, Z., Smoke and Smoked Fish Production. Meat Technology. 2019, 60, 37-43. https://doi.org/10.18485/meattech.2019.60.1.6.
  9. Nieva-Echevarría, B.; Goicoechea, E.; Guillén, M. D., Effect of liquid smoking on lipid hydrolysis and oxidation reactions during in vitro gastrointestinal digestion of European sea bass. Food Research International. 2017, 97, 51-61. https://doi.org/https://doi.org/10.1016/j.foodres.2017.03.032.
  10. Dien, H. A.; Montolalu, R. I.; Berhimpon, S., Liquid smoke inhibits growth of pathogenic and histamine forming bacteria on skipjack fillets. IOP Conference Series: Earth and Environmental Science. 2019, 278, (1), 012018. https://doi.org/10.1088/1755-1315/278/1/012018.
  11. Cuevas, F. J.; Moreno-Rojas, J. M.; Ruiz-Moreno, M. J., Assessing a traceability technique in fresh oranges (Citrus sinensis L. Osbeck) with an HS-SPME-GC-MS method. Towards a volatile characterisation of organic oranges. Food Chemistry. 2017, 221, 1930-1938. https://doi.org/https://doi.org/10.1016/j.foodchem.2016.11.156.
  12. Feng, X.; Ng, V. K.; Mikš-Krajnik, M.; Yang, H., Effects of Fish Gelatin and Tea Polyphenol Coating on the Spoilage and Degradation of Myofibril in Fish Fillet During Cold Storage. Food and Bioprocess Technology. 2017, 10, (1), 89-102. https://doi.org/10.1007/s11947-016-1798-7.
  13. Wang, S.; Chen, H.; Sun, B., Recent progress in food flavor analysis using gas chromatography–ion mobility spectrometry (GC–IMS). Food Chemistry. 2020, 315, 126158. https://doi.org/https://doi.org/10.1016/j.foodchem.2019.126158.
  14. Liu, A.; Zhang, H.; Liu, T.; Gong, P.; Wang, Y.; Wang, H.; Tian, X.; Liu, Q.; Cui, Q.; Xie, X.; Zhang, L.; Yi, H., Aroma classification and flavor characterization of Streptococcus thermophilus fermented milk by HS-GC-IMS and HS-SPME-GC-TOF/MS. Food Bioscience. 2022, 49, 101832. https://doi.org/https://doi.org/10.1016/j.fbio.2022.101832.
  15. Chen, J.; Tao, L.; Zhang, T.; Zhang, J.; Wu, T.; Luan, D.; Ni, L.; Wang, X.; Zhong, J., Effect of four types of thermal processing methods on the aroma profiles of acidity regulator-treated tilapia muscles using E-nose, HS-SPME-GC-MS, and HS-GC-IMS. 2021.

Round 2

Reviewer 2 Report

The authors have made the suggested changes to the manuscript.

In any case, some considerations are necessary.

Line 372: the authors write “at every processing stage was like Moretti et al. studies [59],…..”.  

It is important to underline that Moretti et al. (2016) studied chemical changes and volatiles formation during processing and ripening of a traditional dry salted fish product prepared from landlocked shad (Alosa fallax lacustris). The fish samples were exclusively salted and not smoked. The samples were studied at 9, 40, 70 and 100 days of maturation.

Consequently, the comparison of the compounds found in the samples of the two studies cannot be considered adequate.

Lines 372- 373: the authors report that “….and these compounds can be produced by a variety of microorganisms [44]”.

Reference n. 44 refers to “Vasconi et al. (2015). Fatty Acid Composition of Freshwater Wild Fish in Subalpine Lakes: A Comparative Study”. The authors would probably have wanted to cite another reference. The study of Vasconi et al(2015) does not refer to volatile compounds deriving from microorganisms. Researchers deal exclusively with the fatty acid composition of freshwater fish samples.

COVER LETTER (response 20): the authors, referring to “cooking process”, have reported that “this operation is more suitable for fish that can be eaten directly, such as salmon, etc”.

I ask: What is the meaning of the sentence “more suitable for fish that can be eaten directly…”? Is the fish more suitable from a microbiological or organoleptic point of view?  In any case, it would be appropriate to specify the reason why the samples were cooked.

Generally, it is well known that smoked fish can be directly consumed, without being cooked, such as salmon, trout, etc. This is true for both traditional smoking (hot or cold smoking) and the “smoke flavouring” treatment.

When Good Manufacturing Practices (GMP) and Good Hygienic Practices (GHP) are implemented and the 'cold chain' (application of refrigeration temperatures) is being maintained, no hazard is to be feared (especially biological hazards) for food.

If the authors prefer to keep the cooking phase, it would be better to specify the reason, as above reported

In this way, the profile of volatile substances of samples cooked after “smoke flavouring” would be described. But it would be a “peculiar volatiles profile”, that includes substances derived from cooking.

The authors write “…Çakir et al. [7] also performed the cooking process….”.

As reported in my revision, these authors cooked their samples with a specific purpose: to monitor the color changes of fillets at various stages of two smoking processes and after samples cooking.

In any case, the authors may describe the processing method used for sea bass samples.

In my opinion, this technique (with fish cooking) cannot give a standardized chemical fingerprints of smoke-flavoured sea bass.

Finally, the sentence “these studies may help to further improve the flavor quality of smoke-flavoured fish” (lines 80-81) should be modified including the cooking process.

The latter has a great influence on the volatiles profile of fish samples, but it is not commonly applied in the industry.

For thThe authors have made the suggested changes to the manuscript.

In any case, some considerations are necessary.

Line 372: the authors write “at every processing stage was like Moretti et al. studies [59],…..”.  

It is important to underline that Moretti et al. (2016) studied chemical changes and volatiles formation during processing and ripening of a traditional dry salted fish product prepared from landlocked shad (Alosa fallax lacustris). The fish samples were exclusively salted and not smoked. The samples were studied at 9, 40, 70 and 100 days of maturation.

Consequently, the comparison of the compounds found in the samples of the two studies cannot be considered adequate.

Lines 372- 373: the authors report that “….and these compounds can be produced by a variety of microorganisms [44]”.

Reference n. 44 refers to “Vasconi et al. (2015). Fatty Acid Composition of Freshwater Wild Fish in Subalpine Lakes: A Comparative Study”. The authors would probably have wanted to cite another reference. The study of Vasconi et al(2015) does not refer to volatile compounds deriving from microorganisms. Researchers deal exclusively with the fatty acid composition of freshwater fish samples.

COVER LETTER (response 20): the authors, referring to “cooking process”, have reported that “this operation is more suitable for fish that can be eaten directly, such as salmon, etc”.

I ask: What is the meaning of the sentence “more suitable for fish that can be eaten directly…”? Is the fish more suitable from a microbiological or organoleptic point of view?  In any case, it would be appropriate to specify the reason why the samples were cooked.

Generally, it is well known that smoked fish can be directly consumed, without being cooked, such as salmon, trout, etc. This is true for both traditional smoking (hot or cold smoking) and the “smoke flavouring” treatment.

When Good Manufacturing Practices (GMP) and Good Hygienic Practices (GHP) are implemented and the 'cold chain' (application of refrigeration temperatures) is being maintained, no hazard is to be feared (especially biological hazards) for food.

If the authors prefer to keep the cooking phase, it would be better to specify the reason, as above reported

In this way, the profile of volatile substances of samples cooked after “smoke flavouring” would be described. But it would be a “peculiar volatiles profile”, that includes substances derived from cooking.

The authors write “…Çakir et al. [7] also performed the cooking process….”.

As reported in my revision, these authors cooked their samples with a specific purpose: to monitor the color changes of fillets at various stages of two smoking processes and after samples cooking.

In any case, the authors may describe the processing method used for sea bass samples.

In my opinion, this technique (with fish cooking) cannot give a standardized chemical fingerprints of smoke-flavoured sea bass.

Finally, the sentence “these studies may help to further improve the flavor quality of smoke-flavoured fish” (lines 80-81) should be modified including the cooking process.

The latter has a great influence on the volatiles profile of fish samples, but it is not commonly applied in the industry.

For this reason, it would be advisable for the authors to modify the following sentence: “This study will provide a theoretical basis for improving the quality of smoke-flavoured sea bass products in the future”The authors have made the suggested changes to the manuscript.

In any case, some considerations are necessary.

Line 372: the authors write “at every processing stage was like Moretti et al. studies [59],…..”.  

It is important to underline that Moretti et al. (2016) studied chemical changes and volatiles formation during processing and ripening of a traditional dry salted fish product prepared from landlocked shad (Alosa fallax lacustris). The fish samples were exclusively salted and not smoked. The samples were studied at 9, 40, 70 and 100 days of maturation.

Consequently, the comparison of the compounds found in the samples of the two studies cannot be considered adequate.

Lines 372- 373: the authors report that “….and these compounds can be produced by a variety of microorganisms [44]”.

Reference n. 44 refers to “Vasconi et al. (2015). Fatty Acid Composition of Freshwater Wild Fish in Subalpine Lakes: A Comparative Study”. The authors would probably have wanted to cite another reference. The study of Vasconi et al(2015) does not refer to volatile compounds deriving from microorganisms. Researchers deal exclusively with the fatty acid composition of freshwater fish samples.

COVER LETTER (response 20): the authors, referring to “cooking process”, have reported that “this operation is more suitable for fish that can be eaten directly, such as salmon, etc”.

I ask: What is the meaning of the sentence “more suitable for fish that can be eaten directly…”? Is the fish more suitable from a microbiological or organoleptic point of view?  In any case, it would be appropriate to specify the reason why the samples were cooked.

Generally, it is well known that smoked fish can be directly consumed, without being cooked, such as salmon, trout, etc. This is true for both traditional smoking (hot or cold smoking) and the “smoke flavouring” treatment.

When Good Manufacturing Practices (GMP) and Good Hygienic Practices (GHP) are implemented and the 'cold chain' (application of refrigeration temperatures) is being maintained, no hazard is to be feared (especially biological hazards) for food.

If the authors prefer to keep the cooking phase, it would be better to specify the reason, as above reported

In this way, the profile of volatile substances of samples cooked after “smoke flavouring” would be described. But it would be a “peculiar volatiles profile”, that includes substances derived from cooking.

The authors write “…Çakir et al. [7] also performed the cooking process….”.

As reported in my revision, these authors cooked their samples with a specific purpose: to monitor the color changes of fillets at various stages of two smoking processes and after samples cooking.

In any case, the authors may describe the processing method used for sea bass samples.

In my opinion, this technique (with fish cooking) cannot give a standardized chemical fingerprints of smoke-flavoured sea bass.

Finally, the sentence “these studies may help to further improve the flavor quality of smoke-flavoured fish” (lines 80-81) should be modified including the cooking process.

The latter has a great influence on the volatiles profile of fish samples, but it is not commonly applied in the industry.

For this reason, it would be advisable for the authors to modify the following sentence: “This study will provide a theoretical basis for improving the quality of smoke-flavoured sea bass products in the future”The authors have made the suggested changes to the manuscript.

In any case, some considerations are necessary.

Line 372: the authors write “at every processing stage was like Moretti et al. studies [59],…..”.  

It is important to underline that Moretti et al. (2016) studied chemical changes and volatiles formation during processing and ripening of a traditional dry salted fish product prepared from landlocked shad (Alosa fallax lacustris). The fish samples were exclusively salted and not smoked. The samples were studied at 9, 40, 70 and 100 days of maturation.

Consequently, the comparison of the compounds found in the samples of the two studies cannot be considered adequate.

Lines 372- 373: the authors report that “….and these compounds can be produced by a variety of microorganisms [44]”.

Reference n. 44 refers to “Vasconi et al. (2015). Fatty Acid Composition of Freshwater Wild Fish in Subalpine Lakes: A Comparative Study”. The authors would probably have wanted to cite another reference. The study of Vasconi et al(2015) does not refer to volatile compounds deriving from microorganisms. Researchers deal exclusively with the fatty acid composition of freshwater fish samples.

COVER LETTER (response 20): the authors, referring to “cooking process”, have reported that “this operation is more suitable for fish that can be eaten directly, such as salmon, etc”.

I ask: What is the meaning of the sentence “more suitable for fish that can be eaten directly…”? Is the fish more suitable from a microbiological or organoleptic point of view?  In any case, it would be appropriate to specify the reason why the samples were cooked.

Generally, it is well known that smoked fish can be directly consumed, without being cooked, such as salmon, trout, etc. This is true for both traditional smoking (hot or cold smoking) and the “smoke flavouring” treatment.

When Good Manufacturing Practices (GMP) and Good Hygienic Practices (GHP) are implemented and the 'cold chain' (application of refrigeration temperatures) is being maintained, no hazard is to be feared (especially biological hazards) for food.

If the authors prefer to keep the cooking phase, it would be better to specify the reason, as above reported

In this way, the profile of volatile substances of samples cooked after “smoke flavouring” would be described. But it would be a “peculiar volatiles profile”, that includes substances derived from cooking.

The authors write “…Çakir et al. [7] also performed the cooking process….”.

As reported in my revision, these authors cooked their samples with a specific purpose: to monitor the color changes of fillets at various stages of two smoking processes and after samples cooking.

In any case, the authors may describe the processing method used for sea bass samples.

In my opinion, this technique (with fish cooking) cannot give a standardized chemical fingerprints of smoke-flavoured sea bass.

Finally, the sentence “these studies may help to further improve the flavor quality of smoke-flavoured fish” (lines 80-81) should be modified including the cooking process.

The latter has a great influence on the volatiles profile of fish samples, but it is not commonly applied in the industry.

For this reason, it would be advisable for the authors to modify the following sentence: “This study will provide a theoretical basis for improving the quality of smoke-flavoured sea bass products in the future”The authors have made the suggested changes to the manuscript.

In any case, some considerations are necessary.

Line 372: the authors write “at every processing stage was like Moretti et al. studies [59],…..”.  

It is important to underline that Moretti et al. (2016) studied chemical changes and volatiles formation during processing and ripening of a traditional dry salted fish product prepared from landlocked shad (Alosa fallax lacustris). The fish samples were exclusively salted and not smoked. The samples were studied at 9, 40, 70 and 100 days of maturation.

Consequently, the comparison of the compounds found in the samples of the two studies cannot be considered adequate.

Lines 372- 373: the authors report that “….and these compounds can be produced by a variety of microorganisms [44]”.

Reference n. 44 refers to “Vasconi et al. (2015). Fatty Acid Composition of Freshwater Wild Fish in Subalpine Lakes: A Comparative Study”. The authors would probably have wanted to cite another reference. The study of Vasconi et al(2015) does not refer to volatile compounds deriving from microorganisms. Researchers deal exclusively with the fatty acid composition of freshwater fish samples.

COVER LETTER (response 20): the authors, referring to “cooking process”, have reported that “this operation is more suitable for fish that can be eaten directly, such as salmon, etc”.

I ask: What is the meaning of the sentence “more suitable for fish that can be eaten directly…”? Is the fish more suitable from a microbiological or organoleptic point of view?  In any case, it would be appropriate to specify the reason why the samples were cooked.

Generally, it is well known that smoked fish can be directly consumed, without being cooked, such as salmon, trout, etc. This is true for both traditional smoking (hot or cold smoking) and the “smoke flavouring” treatment.

When Good Manufacturing Practices (GMP) and Good Hygienic Practices (GHP) are implemented and the 'cold chain' (application of refrigeration temperatures) is being maintained, no hazard is to be feared (especially biological hazards) for food.

If the authors prefer to keep the cooking phase, it would be better to specify the reason, as above reported

In this way, the profile of volatile substances of samples cooked after “smoke flavouring” would be described. But it would be a “peculiar volatiles profile”, that includes substances derived from cooking.

The authors write “…Çakir et al. [7] also performed the cooking process….”.

As reported in my revision, these authors cooked their samples with a specific purpose: to monitor the color changes of fillets at various stages of two smoking processes and after samples cooking.

In any case, the authors may describe the processing method used for sea bass samples.

In my opinion, this technique (with fish cooking) cannot give a standardized chemical fingerprints of smoke-flavoured sea bass.

Finally, the sentence “these studies may help to further improve the flavor quality of smoke-flavoured fish” (lines 80-81) should be modified including the cooking process.

The latter has a great influence on the volatiles profile of fish samples, but it is not commonly applied in the industry.

For this reason, it would be advisable for the authors to modify the following sentence: “This study will provide a theoretical basis for improving the quality of smoke-flavoured sea bass products in the future”The authors have made the suggested changes to the manuscript.

In any case, some considerations are necessary.

Line 372: the authors write “at every processing stage was like Moretti et al. studies [59],…..”.  

It is important to underline that Moretti et al. (2016) studied chemical changes and volatiles formation during processing and ripening of a traditional dry salted fish product prepared from landlocked shad (Alosa fallax lacustris). The fish samples were exclusively salted and not smoked. The samples were studied at 9, 40, 70 and 100 days of maturation.

Consequently, the comparison of the compounds found in the samples of the two studies cannot be considered adequate.

Lines 372- 373: the authors report that “….and these compounds can be produced by a variety of microorganisms [44]”.

Reference n. 44 refers to “Vasconi et al. (2015). Fatty Acid Composition of Freshwater Wild Fish in Subalpine Lakes: A Comparative Study”. The authors would probably have wanted to cite another reference. The study of Vasconi et al(2015) does not refer to volatile compounds deriving from microorganisms. Researchers deal exclusively with the fatty acid composition of freshwater fish samples.

COVER LETTER (response 20): the authors, referring to “cooking process”, have reported that “this operation is more suitable for fish that can be eaten directly, such as salmon, etc”.

I ask: What is the meaning of the sentence “more suitable for fish that can be eaten directly…”? Is the fish more suitable from a microbiological or organoleptic point of view?  In any case, it would be appropriate to specify the reason why the samples were cooked.

Generally, it is well known that smoked fish can be directly consumed, without being cooked, such as salmon, trout, etc. This is true for both traditional smoking (hot or cold smoking) and the “smoke flavouring” treatment.

When Good Manufacturing Practices (GMP) and Good Hygienic Practices (GHP) are implemented and the 'cold chain' (application of refrigeration temperatures) is being maintained, no hazard is to be feared (especially biological hazards) for food.

If the authors prefer to keep the cooking phase, it would be better to specify the reason, as above reported

In this way, the profile of volatile substances of samples cooked after “smoke flavouring” would be described. But it would be a “peculiar volatiles profile”, that includes substances derived from cooking.

The authors write “…Çakir et al. [7] also performed the cooking process….”.

As reported in my revision, these authors cooked their samples with a specific purpose: to monitor the color changes of fillets at various stages of two smoking processes and after samples cooking.

In any case, the authors may describe the processing method used for sea bass samples.

In my opinion, this technique (with fish cooking) cannot give a standardized chemical fingerprints of smoke-flavoured sea bass.

Finally, the sentence “these studies may help to further improve the flavor quality of smoke-flavoured fish” (lines 80-81) should be modified including the cooking process.

The latter has a great influence on the volatiles profile of fish samples, but it is not commonly applied in the industry.

For this reason, it would be advisable for the authors to modify the following sentence: “This study will provide a theoretical basis for improving the quality of smoke-flavoured sea bass products in the future”The authors have made the suggested changes to the manuscript.

In any case, some considerations are necessary.

Line 372: the authors write “at every processing stage was like Moretti et al. studies [59],…..”.  

It is important to underline that Moretti et al. (2016) studied chemical changes and volatiles formation during processing and ripening of a traditional dry salted fish product prepared from landlocked shad (Alosa fallax lacustris). The fish samples were exclusively salted and not smoked. The samples were studied at 9, 40, 70 and 100 days of maturation.

Consequently, the comparison of the compounds found in the samples of the two studies cannot be considered adequate.

Lines 372- 373: the authors report that “….and these compounds can be produced by a variety of microorganisms [44]”.

Reference n. 44 refers to “Vasconi et al. (2015). Fatty Acid Composition of Freshwater Wild Fish in Subalpine Lakes: A Comparative Study”. The authors would probably have wanted to cite another reference. The study of Vasconi et al(2015) does not refer to volatile compounds deriving from microorganisms. Researchers deal exclusively with the fatty acid composition of freshwater fish samples.

COVER LETTER (response 20): the authors, referring to “cooking process”, have reported that “this operation is more suitable for fish that can be eaten directly, such as salmon, etc”.

I ask: What is the meaning of the sentence “more suitable for fish that can be eaten directly…”? Is the fish more suitable from a microbiological or organoleptic point of view?  In any case, it would be appropriate to specify the reason why the samples were cooked.

Generally, it is well known that smoked fish can be directly consumed, without being cooked, such as salmon, trout, etc. This is true for both traditional smoking (hot or cold smoking) and the “smoke flavouring” treatment.

When Good Manufacturing Practices (GMP) and Good Hygienic Practices (GHP) are implemented and the 'cold chain' (application of refrigeration temperatures) is being maintained, no hazard is to be feared (especially biological hazards) for food.

If the authors prefer to keep the cooking phase, it would be better to specify the reason, as above reported

In this way, the profile of volatile substances of samples cooked after “smoke flavouring” would be described. But it would be a “peculiar volatiles profile”, that includes substances derived from cooking.

The authors write “…Çakir et al. [7] also performed the cooking process….”.

As reported in my revision, these authors cooked their samples with a specific purpose: to monitor the color changes of fillets at various stages of two smoking processes and after samples cooking.

In any case, the authors may describe the processing method used for sea bass samples.

In my opinion, this technique (with fish cooking) cannot give a standardized chemical fingerprints of smoke-flavoured sea bass.

Finally, the sentence “these studies may help to further improve the flavor quality of smoke-flavoured fish” (lines 80-81) should be modified including the cooking process.

The latter has a great influence on the volatiles profile of fish samples, but it is not commonly applied in the industry.

For this reason, it would be advisable for the authors to modify the following sentence: “This study will provide a theoretical basis for improving the quality of smoke-flavoured sea bass products in the future”The authors have made the suggested changes to the manuscript.

In any case, some considerations are necessary.

Line 372: the authors write “at every processing stage was like Moretti et al. studies [59],…..”.  

It is important to underline that Moretti et al. (2016) studied chemical changes and volatiles formation during processing and ripening of a traditional dry salted fish product prepared from landlocked shad (Alosa fallax lacustris). The fish samples were exclusively salted and not smoked. The samples were studied at 9, 40, 70 and 100 days of maturation.

Consequently, the comparison of the compounds found in the samples of the two studies cannot be considered adequate.

Lines 372- 373: the authors report that “….and these compounds can be produced by a variety of microorganisms [44]”.

Reference n. 44 refers to “Vasconi et al. (2015). Fatty Acid Composition of Freshwater Wild Fish in Subalpine Lakes: A Comparative Study”. The authors would probably have wanted to cite another reference. The study of Vasconi et al(2015) does not refer to volatile compounds deriving from microorganisms. Researchers deal exclusively with the fatty acid composition of freshwater fish samples.

COVER LETTER (response 20): the authors, referring to “cooking process”, have reported that “this operation is more suitable for fish that can be eaten directly, such as salmon, etc”.

I ask: What is the meaning of the sentence “more suitable for fish that can be eaten directly…”? Is the fish more suitable from a microbiological or organoleptic point of view?  In any case, it would be appropriate to specify the reason why the samples were cooked.

Generally, it is well known that smoked fish can be directly consumed, without being cooked, such as salmon, trout, etc. This is true for both traditional smoking (hot or cold smoking) and the “smoke flavouring” treatment.

When Good Manufacturing Practices (GMP) and Good Hygienic Practices (GHP) are implemented and the 'cold chain' (application of refrigeration temperatures) is being maintained, no hazard is to be feared (especially biological hazards) for food.

If the authors prefer to keep the cooking phase, it would be better to specify the reason, as above reported

In this way, the profile of volatile substances of samples cooked after “smoke flavouring” would be described. But it would be a “peculiar volatiles profile”, that includes substances derived from cooking.

The authors write “…Çakir et al. [7] also performed the cooking process….”.

As reported in my revision, these authors cooked their samples with a specific purpose: to monitor the color changes of fillets at various stages of two smoking processes and after samples cooking.

In any case, the authors may describe the processing method used for sea bass samples.

In my opinion, this technique (with fish cooking) cannot give a standardized chemical fingerprints of smoke-flavoured sea bass.

Finally, the sentence “these studies may help to further improve the flavor quality of smoke-flavoured fish” (lines 80-81) should be modified including the cooking process.

The latter has a great influence on the volatiles profile of fish samples, but it is not commonly applied in the industry.

For this reason, it would be advisable for the authors to modify the following sentence: “This study will provide a theoretical basis for improving the quality of smoke-flavoured sea bass products in the future”The authors have made the suggested changes to the manuscript.

In any case, some considerations are necessary.

Line 372: the authors write “at every processing stage was like Moretti et al. studies [59],…..”.  

It is important to underline that Moretti et al. (2016) studied chemical changes and volatiles formation during processing and ripening of a traditional dry salted fish product prepared from landlocked shad (Alosa fallax lacustris). The fish samples were exclusively salted and not smoked. The samples were studied at 9, 40, 70 and 100 days of maturation.

Consequently, the comparison of the compounds found in the samples of the two studies cannot be considered adequate.

Lines 372- 373: the authors report that “….and these compounds can be produced by a variety of microorganisms [44]”.

Reference n. 44 refers to “Vasconi et al. (2015). Fatty Acid Composition of Freshwater Wild Fish in Subalpine Lakes: A Comparative Study”. The authors would probably have wanted to cite another reference. The study of Vasconi et al(2015) does not refer to volatile compounds deriving from microorganisms. Researchers deal exclusively with the fatty acid composition of freshwater fish samples.

COVER LETTER (response 20): the authors, referring to “cooking process”, have reported that “this operation is more suitable for fish that can be eaten directly, such as salmon, etc”.

I ask: What is the meaning of the sentence “more suitable for fish that can be eaten directly…”? Is the fish more suitable from a microbiological or organoleptic point of view?  In any case, it would be appropriate to specify the reason why the samples were cooked.

Generally, it is well known that smoked fish can be directly consumed, without being cooked, such as salmon, trout, etc. This is true for both traditional smoking (hot or cold smoking) and the “smoke flavouring” treatment.

When Good Manufacturing Practices (GMP) and Good Hygienic Practices (GHP) are implemented and the 'cold chain' (application of refrigeration temperatures) is being maintained, no hazard is to be feared (especially biological hazards) for food.

If the authors prefer to keep the cooking phase, it would be better to specify the reason, as above reported

In this way, the profile of volatile substances of samples cooked after “smoke flavouring” would be described. But it would be a “peculiar volatiles profile”, that includes substances derived from cooking.

The authors write “…Çakir et al. [7] also performed the cooking process….”.

As reported in my revision, these authors cooked their samples with a specific purpose: to monitor the color changes of fillets at various stages of two smoking processes and after samples cooking.

In any case, the authors may describe the processing method used for sea bass samples.

In my opinion, this technique (with fish cooking) cannot give a standardized chemical fingerprints of smoke-flavoured sea bass.

Finally, the sentence “these studies may help to further improve the flavor quality of smoke-flavoured fish” (lines 80-81) should be modified including the cooking process.

The latter has a great influence on the volatiles profile of fish samples, but it is not commonly applied in the industry.

For this reason, it would be advisable for the authors to modify the following sentence: “This study will provide a theoretical basis for improving the quality of smoke-flavoured sea bass products in the future”The authors have made the suggested changes to the manuscript.

In any case, some considerations are necessary.

Line 372: the authors write “at every processing stage was like Moretti et al. studies [59],…..”.  

It is important to underline that Moretti et al. (2016) studied chemical changes and volatiles formation during processing and ripening of a traditional dry salted fish product prepared from landlocked shad (Alosa fallax lacustris). The fish samples were exclusively salted and not smoked. The samples were studied at 9, 40, 70 and 100 days of maturation.

Consequently, the comparison of the compounds found in the samples of the two studies cannot be considered adequate.

Lines 372- 373: the authors report that “….and these compounds can be produced by a variety of microorganisms [44]”.

Reference n. 44 refers to “Vasconi et al. (2015). Fatty Acid Composition of Freshwater Wild Fish in Subalpine Lakes: A Comparative Study”. The authors would probably have wanted to cite another reference. The study of Vasconi et al(2015) does not refer to volatile compounds deriving from microorganisms. Researchers deal exclusively with the fatty acid composition of freshwater fish samples.

COVER LETTER (response 20): the authors, referring to “cooking process”, have reported that “this operation is more suitable for fish that can be eaten directly, such as salmon, etc”.

I ask: What is the meaning of the sentence “more suitable for fish that can be eaten directly…”? Is the fish more suitable from a microbiological or organoleptic point of view?  In any case, it would be appropriate to specify the reason why the samples were cooked.

Generally, it is well known that smoked fish can be directly consumed, without being cooked, such as salmon, trout, etc. This is true for both traditional smoking (hot or cold smoking) and the “smoke flavouring” treatment.

When Good Manufacturing Practices (GMP) and Good Hygienic Practices (GHP) are implemented and the 'cold chain' (application of refrigeration temperatures) is being maintained, no hazard is to be feared (especially biological hazards) for food.

If the authors prefer to keep the cooking phase, it would be better to specify the reason, as above reported

In this way, the profile of volatile substances of samples cooked after “smoke flavouring” would be described. But it would be a “peculiar volatiles profile”, that includes substances derived from cooking.

The authors write “…Çakir et al. [7] also performed the cooking process….”.

As reported in my revision, these authors cooked their samples with a specific purpose: to monitor the color changes of fillets at various stages of two smoking processes and after samples cooking.

In any case, the authors may describe the processing method used for sea bass samples.

In my opinion, this technique (with fish cooking) cannot give a standardized chemical fingerprints of smoke-flavoured sea bass.

Finally, the sentence “these studies may help to further improve the flavor quality of smoke-flavoured fish” (lines 80-81) should be modified including the cooking process.

The latter has a great influence on the volatiles profile of fish samples, but it is not commonly applied in the industry.

For this reason, it would be advisable for the authors to modify the following sentence: “This study will provide a theoretical basis for improving the quality of smoke-flavoured sea bass products in the future”The authors have made the suggested changes to the manuscript.

In any case, some considerations are necessary.

Line 372: the authors write “at every processing stage was like Moretti et al. studies [59],…..”.  

It is important to underline that Moretti et al. (2016) studied chemical changes and volatiles formation during processing and ripening of a traditional dry salted fish product prepared from landlocked shad (Alosa fallax lacustris). The fish samples were exclusively salted and not smoked. The samples were studied at 9, 40, 70 and 100 days of maturation.

Consequently, the comparison of the compounds found in the samples of the two studies cannot be considered adequate.

Lines 372- 373: the authors report that “….and these compounds can be produced by a variety of microorganisms [44]”.

Reference n. 44 refers to “Vasconi et al. (2015). Fatty Acid Composition of Freshwater Wild Fish in Subalpine Lakes: A Comparative Study”. The authors would probably have wanted to cite another reference. The study of Vasconi et al(2015) does not refer to volatile compounds deriving from microorganisms. Researchers deal exclusively with the fatty acid composition of freshwater fish samples.

COVER LETTER (response 20): the authors, referring to “cooking process”, have reported that “this operation is more suitable for fish that can be eaten directly, such as salmon, etc”.

I ask: What is the meaning of the sentence “more suitable for fish that can be eaten directly…”? Is the fish more suitable from a microbiological or organoleptic point of view?  In any case, it would be appropriate to specify the reason why the samples were cooked.

Generally, it is well known that smoked fish can be directly consumed, without being cooked, such as salmon, trout, etc. This is true for both traditional smoking (hot or cold smoking) and the “smoke flavouring” treatment.

When Good Manufacturing Practices (GMP) and Good Hygienic Practices (GHP) are implemented and the 'cold chain' (application of refrigeration temperatures) is being maintained, no hazard is to be feared (especially biological hazards) for food.

If the authors prefer to keep the cooking phase, it would be better to specify the reason, as above reported

In this way, the profile of volatile substances of samples cooked after “smoke flavouring” would be described. But it would be a “peculiar volatiles profile”, that includes substances derived from cooking.

The authors write “…Çakir et al. [7] also performed the cooking process….”.

As reported in my revision, these authors cooked their samples with a specific purpose: to monitor the color changes of fillets at various stages of two smoking processes and after samples cooking.

In any case, the authors may describe the processing method used for sea bass samples.

In my opinion, this technique (with fish cooking) cannot give a standardized chemical fingerprints of smoke-flavoured sea bass.

Finally, the sentence “these studies may help to further improve the flavor quality of smoke-flavoured fish” (lines 80-81) should be modified including the cooking process.

The latter has a great influence on the volatiles profile of fish samples, but it is not commonly applied in the industry.

For this reason, it would be advisable for the authors to modify the following sentence: “This study will provide a theoretical basis for improving the quality of smoke-flavoured sea bass products in the future”The authors have made the suggested changes to the manuscript.

In any case, some considerations are necessary.

Line 372: the authors write “at every processing stage was like Moretti et al. studies [59],…..”.  

It is important to underline that Moretti et al. (2016) studied chemical changes and volatiles formation during processing and ripening of a traditional dry salted fish product prepared from landlocked shad (Alosa fallax lacustris). The fish samples were exclusively salted and not smoked. The samples were studied at 9, 40, 70 and 100 days of maturation.

Consequently, the comparison of the compounds found in the samples of the two studies cannot be considered adequate.

Lines 372- 373: the authors report that “….and these compounds can be produced by a variety of microorganisms [44]”.

Reference n. 44 refers to “Vasconi et al. (2015). Fatty Acid Composition of Freshwater Wild Fish in Subalpine Lakes: A Comparative Study”. The authors would probably have wanted to cite another reference. The study of Vasconi et al(2015) does not refer to volatile compounds deriving from microorganisms. Researchers deal exclusively with the fatty acid composition of freshwater fish samples.

COVER LETTER (response 20): the authors, referring to “cooking process”, have reported that “this operation is more suitable for fish that can be eaten directly, such as salmon, etc”.

I ask: What is the meaning of the sentence “more suitable for fish that can be eaten directly…”? Is the fish more suitable from a microbiological or organoleptic point of view?  In any case, it would be appropriate to specify the reason why the samples were cooked.

Generally, it is well known that smoked fish can be directly consumed, without being cooked, such as salmon, trout, etc. This is true for both traditional smoking (hot or cold smoking) and the “smoke flavouring” treatment.

When Good Manufacturing Practices (GMP) and Good Hygienic Practices (GHP) are implemented and the 'cold chain' (application of refrigeration temperatures) is being maintained, no hazard is to be feared (especially biological hazards) for food.

If the authors prefer to keep the cooking phase, it would be better to specify the reason, as above reported

In this way, the profile of volatile substances of samples cooked after “smoke flavouring” would be described. But it would be a “peculiar volatiles profile”, that includes substances derived from cooking.

The authors write “…Çakir et al. [7] also performed the cooking process….”.

As reported in my revision, these authors cooked their samples with a specific purpose: to monitor the color changes of fillets at various stages of two smoking processes and after samples cooking.

In any case, the authors may describe the processing method used for sea bass samples.

In my opinion, this technique (with fish cooking) cannot give a standardized chemical fingerprints of smoke-flavoured sea bass.

Finally, the sentence “these studies may help to further improve the flavor quality of smoke-flavoured fish” (lines 80-81) should be modified including the cooking process.

The latter has a great influence on the volatiles profile of fish samples, but it is not commonly applied in the industry.

For this reason, it would be advisable for the authors to modify the following sentence: “This study will provide a theoretical basis for improving the quality of smoke-flavoured sea bass products in the future”is reason, it would be advisable for the authors to modify the following sentence: “This study will provide a theoretical basis for improving the quality of smoke-flavoured sea bass products in the future”

Author Response

Response to Reviewer 2 Comments

Point 1: The authors have made the suggested changes to the manuscript. In any case, some considerations are necessary.

Response 1: We are extremely grateful to the respected reviewer for the positive comments and helpful suggestions. We have carefully revised the manuscript according to your comments. We assume that the paper has been enhanced and we hope that the corrections made, along with the reply attached, will address the concerns raised. Below you will find our point-by-point responses to the comments/ questions addressed.

Point 2: Line 372: the authors write “at every processing stage was like Moretti et al. studies [59],…..”. 

It is important to underline that Moretti et al. (2016) studied chemical changes and volatiles formation during processing and ripening of a traditional dry salted fish product prepared from landlocked shad (Alosa fallax lacustris). The fish samples were exclusively salted and not smoked. The samples were studied at 9, 40, 70 and 100 days of maturation.

Consequently, the comparison of the compounds found in the samples of the two studies cannot be considered adequate.

Response 2: Thanks for the clarification. As you suggest, the comparison of the compounds found in the samples of the two studies cannot be considered adequate. After informative references, Varlet et al. [1] showed that the important quantities of n-alkanals found in smoked fish flesh could be related to the important amount of their lipidic precursors found in unsmoked fish flesh[2]. The corresponding part has been rewritten in our revised manuscript. Please check it.

Point 3: Lines 372- 373: the authors report that “….and these compounds can be produced by a variety of microorganisms [44]”.

Reference n. 44 refers to “Vasconi et al. (2015). Fatty Acid Composition of Freshwater Wild Fish in Subalpine Lakes: A Comparative Study”. The authors would probably have wanted to cite another reference. The study of Vasconi et al(2015) does not refer to volatile compounds deriving from microorganisms. Researchers deal exclusively with the fatty acid composition of freshwater fish samples.

Response 3: Thanks for your careful observation. We feel very sorry for a mistake of our jobs. In fact, "….and these compounds can be produced by a variety of microorganisms" refers to the report of MORETTI et al.[3] "The most abundant were pentadecane, pentadecene, heptadecane and heptadecene. They can be produced by a wide range of microorganisms and freshwater cyanobacteria (Shakeel et al. 2015)." However, we found in the study of Shakeel et al.[4] that they selected cyanobacterial isolates from Indian marine and freshwater habitats and screened them for the production of alkanes and alkenes. Meanwhile, alkane production is thought to proceed primarily through the fatty acid synthesis (FAS) pathway followed by reduction and a carbonyl group removal via AAR and ADO enzymes[5]. Therefore, our interpretation of "….and these compounds can be produced by a variety of microorganisms "in this way may be uncertain. We decided to delete this sentence in the manuscript in order not to affect the reader's identification.

Point 4: COVER LETTER (response 20): the authors, referring to “cooking process”, have reported that “this operation is more suitable for fish that can be eaten directly, such as salmon, etc”.

I ask: What is the meaning of the sentence “more suitable for fish that can be eaten directly…”? Is the fish more suitable from a microbiological or organoleptic point of view?  In any case, it would be appropriate to specify the reason why the samples were cooked.

Generally, it is well known that smoked fish can be directly consumed, without being cooked, such as salmon, trout, etc. This is true for both traditional smoking (hot or cold smoking) and the “smoke flavouring” treatment.

When Good Manufacturing Practices (GMP) and Good Hygienic Practices (GHP) are implemented and the 'cold chain' (application of refrigeration temperatures) is being maintained, no hazard is to be feared (especially biological hazards) for food.

If the authors prefer to keep the cooking phase, it would be better to specify the reason, as above reported

In this way, the profile of volatile substances of samples cooked after “smoke flavouring” would be described. But it would be a “peculiar volatiles profile”, that includes substances derived from cooking.

The authors write “…Çakir et al. [7] also performed the cooking process….”.

As reported in my revision, these authors cooked their samples with a specific purpose: to monitor the color changes of fillets at various stages of two smoking processes and after samples cooking.

In any case, the authors may describe the processing method used for sea bass samples.

In my opinion, this technique (with fish cooking) cannot give a standardized chemical fingerprints of smoke-flavoured sea bass.

Response 4: Thanks for your careful observation and valuable advice. It has been well documented that smoked fish can be directly consumed, without being cooked, such as salmon, trout, etc. This is true for both traditional smoking (hot or cold smoking) and the “smoke flavouring” treatment. However, brining, drying, heating, and the reactivity of smoke components may have an impact on the rate of lipid changes by affecting the tissue enzymes involved in oxidation reactions, as well as by generating and changing the stability of radicals.[6] In addition, when sea bass fillets are treated with smoke flavorings, the first important change is the incorporation on to the fillets of both a numerous group of smoke components, as well as the carrier in which they are supported. In turn, the addition of these components leads to a change in sensory characteristics. So based on the best sensory characteristics of smoke-flavored sea bass, we also did a lot of experiments in our previous work. Of course, the sensory impact of smoke flavors is at least as important as the preservative effect, which is why we finally decided to keep the cooking stage. For the relationship between this technique (with fish cooking) and the standardized chemical fingerprints of smoke-flavoured sea bass, it has been rephrased now in the revised manuscript. We hope to receive your approval in our response. Thank you again for your valuable comments.

Point 5: Finally, the sentence “these studies may help to further improve the flavor quality of smoke-flavoured fish” (lines 80-81) should be modified including the cooking process.

The latter has a great influence on the volatiles profile of fish samples, but it is not commonly applied in the industry.

Response 5: Thanks for your valuable remark. It has been changed now as mentioned.

References

  1. Varlet, V.; Prost, C.; Serot, T., Volatile aldehydes in smoked fish: Analysis methods, occurence and mechanisms of formation. Food Chemistry. 2007, 105, (4), 1536-1556. https://doi.org/https://doi.org/10.1016/j.foodchem.2007.03.041.
  2. Varlet, V.; Knockaert, C.; Prost, C.; Serot, T., Comparison of Odor-Active Volatile Compounds of Fresh and Smoked Salmon. Journal of Agricultural and Food Chemistry. 2006, 54, (9), 3391-3401. https://doi.org/10.1021/jf053001p.
  3. Moretti, V. M.; Vasconi, M.; Caprino, F.; Bellagamba, F., Fatty Acid Profiles and Volatile Compounds Formation During Processing and Ripening of a Traditional Salted Dry Fish Product. Journal of Food Processing & Preservation. 2016, 41.
  4. Shakeel, T.; Fatma, Z.; Fatma, T.; Yazdani, S. S., Heterogeneity of Alkane Chain Length in Freshwater and Marine Cyanobacteria. Frontiers in Bioengineering and Biotechnology. 2015, 3. https://doi.org/10.3389/fbioe.2015.00034.
  5. Schirmer, A.; Rude, M. A.; Li, X.; Popova, E.; del Cardayre, S. B., Microbial Biosynthesis of Alkanes. Science. 2010, 329, (5991), 559-562. https://doi.org/10.1126/science.1187936.
  6. Stołyhwo, A.; Kołodziejska, I.; Sikorski, Z. E., Long chain polyunsaturated fatty acids in smoked Atlantic mackerel and Baltic sprats. Food Chemistry. 2006, 94, (4), 589-595. https://doi.org/https://doi.org/10.1016/j.foodchem.2004.11.050.
